# Quantifying Trunk Impact Dynamics and Workload with Inertial Sensors in Goalball Players

**DOI:** 10.3390/sports12110291

**Published:** 2024-10-24

**Authors:** Cristina Comeras-Chueca, Pablo J. Bascuas, César Berzosa, Eduardo Piedrafita, Juan Rabal-Pelay, Héctor Gutiérrez, Ana Vanessa Bataller-Cervero

**Affiliations:** 1Health Sciences Faculty, Universidad San Jorge, Autov. A-23 km 299, Villanueva de Gállego, 50830 Zaragoza, Spain; ccomeras@usj.es (C.C.-C.); cberzosa@usj.es (C.B.); epiedrafita@usj.es (E.P.); jrabal@usj.es (J.R.-P.); hgutierrez@usj.es (H.G.); avbataller@usj.es (A.V.B.-C.); 2ValorA Research Group, Health Sciences Faculty, Universidad San Jorge, 50830 Zaragoza, Spain

**Keywords:** paralympic sports, wearable sensor, accelerometry, dynamic stress load, pilot study

## Abstract

The aim of this study was to quantify trunk impact dynamics in goalball players using inertial sensors and evaluate the goalball players’ workloads, focusing on changes between the first and second halves of a match to enhance understanding of the demands experienced throughout the game. Utilizing inertial technology, trunk impacts during goalball gameplay were analyzed to provide a holistic insight into how these impacts influence athletes’ dynamic stress loads, which refers to the physical demands placed on the body during dynamic movements. Six goalball players were recruited to wear an accelerometer during a whole goalball game to quantify trunk impacts. The results showed a higher number of total impacts and a higher number of impacts at higher levels during the first half, compared to a higher percentage of impacts at a low impact level in the second half. These results suggest that the intensity of gameplay is related to the number of impacts sustained, with players experiencing significantly more impacts, particularly at very low, low, and very high impact levels, in the first half compared to the second half. This decline in impacts aligns with the reduction in game intensity as the match progressed, as indicated by a lower heart rate and a trend towards a lower dynamic stress load in the second half. Future research could explore targeted training interventions aimed at optimizing workload and performance in goalball players.

## 1. Introduction

The landscape of sports for individuals with visual impairments has garnered significant attention in recent research. Goalball, established as a Paralympic sport in Toronto in 1976, involves two teams of three visually impaired players, who also wear opaque eye goggles to ensure uniformity, competing on a volleyball court [1]. The game employs an acoustic sensory ball, and the objective is to score by propelling it into the opposing team’s goal line. Matches, with a 24 min total duration, consist of two 12 min halves, potentially concluding early if one team attains a 10-goal lead [2].

In goalball, players seamlessly transition between offense and defense, each role demanding distinct performance factors [2]. While attacking emphasizes throwing velocity, precision, and balance, defending involves players preventing the ball from crossing the goal-line by throwing themselves to the ground to block the ball [3], and these impacts could contribute to increased fatigue. However, there is a lack of evidence concerning the demands faced and how to assess them in goalball [3].

Crucial factors that can influence the performance of the goalball players include upper and lower limb strength, as well as core stability [4]. Core muscles, essential for spine stability, positively impact overhead throwing and velocity [5], and may also play a crucial role during defensive impacts, potentially contributing to the workload encountered. Core training has been found to enhance balance in visually impaired individuals [6]. Thus, enhanced core strength, and consequently improved balance, could also positively impact the defensive phase, reducing workload and potential injuries resulting from impacts when falling to the ground to intercept opponent throws.

Regarding game workloads and fatigue in goalball, Cursiol et al. [7] showed the occurrence of fatigue in the second and third games compared to the first game on the same day. It is important to highlight that, according to the Goalball Rules and Regulations for the 2022–2024 period issued by the International Blind Sports Federation [2], a goalball team can play up to three games in the same day. Moreover, the frequency of throwing themselves to the ground to intercept throws during defense, as well as the players’ reaction time, could also be influenced by the workload. Quantifying the external load from impacts on players would be helpful; this can be assessed using various technological devices that not only quantify the impacts received but also classify them by intensity [8].

The quantification of impacts during the defensive phase on goalball players holds significant potential for optimizing the performance of visually impaired players. In the context of sports physiology, it is well established that repeated impacts can contribute to increased workload and exercise-induced fatigue, thereby potentially increasing the overall stress load on players’ bodies [9]. A comprehensive understanding, derived from measuring the impacts experienced by goalball players, not only can enhance sports performance but also contributes to the development of tailored recovery protocols. Regarding this aspect, inertial sensors in sports monitoring have revolutionized the way external training load is quantified [10], including in Paralympic sports [8]. The integration of inertial systems enables a more comprehensive evaluation of workload, particularly in the context of dynamic stress load (DSL), which measures the mechanical stress experienced by players during exercise, considering the intensity and variability of movement patterns during training or competition [8,10].

This exploratory pilot study aims to utilize inertial technology to quantify the dynamics of trunk impacts in goalball players, as well as to research the changes in the number and intensity of trunk impacts between the first and second halves of a goalball match. By analyzing these impacts, the study seeks to elucidate how they can reflect the progression of DSL throughout the game. The hypothesis of this study was that trunk impact dynamics change between the first and second halves of a goalball match due to the workload from the impacts in the first half, and that players modify their defensive strategies to avoid extended exposure to high-intensity impacts, choosing a more conservative approach to save energy. This reduction in impacts during the second half could also be attributed to the increasing workload as the game progresses, which may influence players’ ability to maintain the same intensity. While previous research has explored fatigue across multiple games, there is a need to better understand how workload affects impact dynamics within individual games, particularly between halves.

## 2. Materials and Methods

### 2.1. Study Design

This research is an exploratory pilot study that assessed the impacts sustained by players during a national league goalball match. Informed consent forms were signed by players after the procedures were explained and before study commencement with the collaboration of the researchers and the National Organization of Spanish Blind People (Organización Nacional de Ciegos Españoles: ONCE) collaborators. In addition, personal data were collected with a Personal Questionnaire Form (i.e., age, degree of disability, goalball training experience, injuries suffered). It is worth noting that the information sheet about the study and the informed consent form were read aloud to the players in the presence of the coach, and subsequently, they either signed the informed consent form or provided verbal consent, in the case of being unable to write.

This study was performed in accordance with the ethical guidelines of the Helsinki Declaration of 1964 (revised in Fortaleza, 2013). The study has been reviewed and approved by the Ethics Committee of the University of San Jorge (code 06/1/21-22).

### 2.2. Participants

Six goalball players (32 ± 8 years old; 4 males and 2 females) participated in this study. The eligibility criteria were as follows: (1) goalball players affiliated with the Spanish Federation of Sports for Blinded People (Federación Española de Deportes para Ciegos); and (2) actively participating in the highest category during the 2021–2022 season. Individuals who experienced neuromuscular or cardiovascular conditions within the three months preceding the study were not included. Participants were recruited from a goalball team associated with ONCE. Participants had a mean body mass of 76.5 ± 20.1 kg and a height of 172 ± 8 cm. On average, they had 10 ± 2 years of experience in playing goalball.

### 2.3. Equipment

A STATSports Apex (STATSports Group; Newry, UK) device incorporates three-axis accelerometer technology with a sampling rate of 100 Hz, enabling the detection of maximum accelerometer impacts of 2 g within a 0.01 s period. In this research, accelerometers were strategically positioned on the upper back, and the analysis focused on the percentage of impacts categorized by positive and negative slopes. These accelerometers were housed within a fabric harness with a pocket located around the first thoracic vertebrae. It is noteworthy that, being padded, the harness did not pose any issues for the players in the event of falls. In addition to the factory calibration, prior to each session, the sensors were reset to a horizontal position on a flat surface to ensure accurate measurements. This ensured that, at rest, the acceleration values across all three axes were 0 m/s^2^, providing a reliable baseline for measuring impacts during gameplay. This additional step helped verify the accuracy of the impact data collected from the players.

The impacts were sorted into six zones or levels: very low impact level (impacts between 3 and 4 arbitrary units (AU)), low impact level (impacts between 4 and 5 AU), moderate impact level (impacts between 5 and 6 AU), high impact level (impacts between 6 and 7 AU), very high impact level (impacts between 7 and 8 AU), and extreme impact level (impacts between 8 and 9 AU). Impacts below 3 AU and above 9 AU were excluded from consideration. Subsequently, all data were collected and organized using Excel^®^ (2016, Microsoft, Inc., Redmond, WA, USA) for subsequent statistical analysis.

The rationale for setting the interval between 3 AU and 9 AU is based on the nature and occurrence of impacts in sports. Impacts above 9 AU are excluded because they are not typically encountered in sports and they are considered outliers, and this is also the measurement limit; the inertial sensors apply a filter, and do not calculate them. In fact, the maximum recorded impact in the present study is 8.3 AU. Impacts below 3 AU are excluded as well, as they likely pose minimal injury risk and are similar to the AU experienced in daily life. The decision to exclude impacts less than 3 AU and greater than 9 AU was based on their limited relevance to the specific context of the sport being studied, where impacts within this range are most representative of the physical demands placed on players.

The categories for impact severity (very low, low, moderate, high, very high, and extreme) were chosen based on a combination of factors, including existing literature on similar sports [11], recommendations from device manufacturers, and the unique context of this study.

We recorded data only during the first and second halves of the game, excluding warm-up and cool-down activities, and any stoppages. The analysis focused solely on active gameplay time, ensuring that only the actual demands placed on the players during the game were measured. All players participated for the entirety of each half, and the duration of the halves was consistent with the standard rules of goalball.

### 2.4. Dynamic Stress Load (DSL)

DSL is a variable related to the player workload during exercise [8,10]. DSL refers to the magnitude of linear acceleration experienced by the player during exercise and is calculated using a 100 Hz accelerometer measuring linear accelerations in the three axes of movement (X, Y, Z). Acceleration values were combined over the specified timeframe to derive a metric for DSL using the following equation for DSL calculation:$$DSL=\sum_{i=1}^{n}{IMPACT}_{i}^{k}\cdot SF$$
where *K* represents a weighting factor, and SF stands for a scaling factor. *IMPACT* is characterized as the highest composite acceleration value among every set of 10 accelerometer points recorded, given that this maximum value exceeds 3G. These factors are undisclosed proprietary values.

Previous studies have demonstrated that the reliability of accelerometer data, including DSL, is acceptable during various activities such as running and high-intensity interval training. Specifically, the reliability of DSL has been shown to have a coefficient of variation of less than 10% when monitoring external loads in soccer training sessions [12]. This indicates that DSL can be a reliable metric for assessing the dynamic load experienced by athletes during these activities.

DSL, as a metric that quantifies the external mechanical load experienced by an athlete during exercise [13,14], is particularly important in activities involving repetitive or sustained mechanical loads, like the impacts experienced when you throw yourself to the ground during the defensive phase in goalball, where maintaining performance is closely tied to how well the body manages and distributes these loads to delay the onset of fatigue [14].

### 2.5. Heart Rate

Heart rate was continuously monitored throughout the study using Garmin heart rate bands synchronized with the inertial sensors always worn by the players. This allowed for real-time tracking of heart rate fluctuations during exercise sessions. The data collected from the heart rate bands provided valuable insights into the cardiovascular response of the players to various intensities of physical activity and environmental conditions. Furthermore, the integration of heart rate monitoring with inertial sensors facilitated a comprehensive understanding of the relationship between physiological responses and performance metrics, enhancing the overall analysis of the study.

### 2.6. Statistical Analysis

The sampling employed in this study was purposeful, rendering the calculation of the sample size inappropriate. Instead, probabilistic sampling techniques were utilized for this calculation. While this approach imposes limitations on external validity and the generalizability of results, it does not compromise the internal validity of the findings. Due to the limited availability of players who meet the specific inclusion criteria, the small sample size highlights the challenges in recruiting a larger group, making it a necessary step for conducting initial exploratory analysis in this under-researched sport.

Statistical analyses were conducted using the Statistical Package for the Social Sciences (SPSS) version 29.0 (SPSS Inc., Chicago, IL, USA). The normal distribution of variables was verified through Kolmogorov–Smirnov tests, revealing that several variables did not adhere to a normal distribution. Given the non-normal distribution of the data, non-parametric tests were chosen, as they do not assume normality and are more robust against such deviations. This choice ensures that the analyses remain valid and reliable, particularly in studies involving small sample sizes or ordinal data. Consequently, non-parametric tests were employed. Significance levels were set at *p* < 0.05 for all tests, and data are presented as medians and interquartile ranges.

Differences between the first and second halves of the game were explored using a Wilcoxon signed-rank test. The Wilcoxon test was chosen given the non-normal distribution of the data and its appropriateness for comparing paired samples. This analysis allowed for the examination of potential changes or shifts in variables across different phases of the game, providing valuable insights into the dynamics throughout the game. Effect size (ES) was calculated as $ES=\frac{W}{\sqrt{n}}$ using the criteria of ES < 0.1 as trivial, 0.1 ≤ ES < 0.3 as small, 0.3 ≤ ES < 0.5 as medium, and ES ≥ 0.5 as large [15]. Here, *W* represents the Wilcoxon signed-rank test statistic, a non-parametric statistical test used to compare two related samples or repeated measurements on a single sample, assessing whether their population mean ranks differ. The *n* denotes the number of observations or pairs of data points in the sample. Specifically, in the context of this research, the *n* refers to the number of goalball players or the number of impact measurements taken during the study. Each player wore a device with a sampling rate of 100 Hz, resulting in multiple impact measurements for each player across games. This approach allows for a robust assessment of the effect size, accounting for the non-parametric nature of the data and the sample size involved.

## 3. Results

The players experienced a median of 104 impacts in total, with a median of 60 impacts at the very low impact level, 21.5 at the low level, 8 at the moderate impact level, 5.5 at the high impact level, 3.5 at the very high impact level, and 3 at the extreme impact level. The number of impacts at each level during the first and second halves can be seen in Table 1, which shows that players experienced significantly more impacts in the first half compared to the second half of the game, both in the total number of impacts and in the number of impacts at very low, low, and very high impact levels.

Descriptive data and statistical differences between DSL, heart rate, and proportions of impacts in each intensity level are presented in Table 2. The results revealed a greater percentage of impacts at a very high impact level, along with a higher maximum heart rate during the first half. In contrast, the second half showed a higher percentage of low impact levels. Those results can be seen in Figure 1, which displays the number of impacts in the first and second halves of the game per individual, both in different impact levels and in the total number of impacts, showing a clear reduction in the number of impacts per participant in the second half, further emphasizing the decline in intensity as the game progressed.

## 4. Discussion

The results of the study on quantifying trunk impact dynamics in goalball players provide valuable insights into the physical demands for visually impaired goalball players. In this study, researchers observed that players experiencing accumulated fatigue may adapt their defensive strategies to minimize prolonged exposure to high-intensity impacts, opting for a more conservative approach to conserve energy. This is supported by evidence indicating fluctuations in goalball players’ performance based on the frequency of their offensive and defensive actions [16]. Such fluctuations can be influenced by the players’ abilities to maintain effort throughout the game [17], as has been seen in the present study. This accumulation of fatigue can lead to a reduction in the muscles’ capacity to generate force, typically resulting from prolonged motor tasks and consequently decreasing voluntary muscle activation [18]. As a result, more impacts, and specifically more impacts at very low and very high intensity, were observed in the first half of the game compared to the second half.

Regarding the analysis of differences between the first and second halves of the game, the results shed light on important aspects of player performance and physiological responses over the course of the game. The results indicate significant disparities between the two halves, notably in the total number of impacts, where a lower number was recorded during the second half compared to the first. This observation suggests a potential effect of workload or tactical adjustments as the game progresses. Additionally, regarding specific impact levels, the findings reveal variations in the distribution of impacts between the halves, with a greater number of impacts in very low and very high impact levels during the first half, and a higher percentage of impacts at the very high impact level and lower percentage of impacts at the low impact level in the second half. These differences may reflect changes in gameplay strategies or defensive tactics as players adapt to the evolving dynamics of the game. Furthermore, while the *p*-values associated with these differences vary, the effect sizes provide additional insights into the practical significance of these findings. For instance, large effect sizes were observed for variables such as the total number of impacts and the maximum heart rate, indicating substantial practical significance with statistically significant differences between the first and the second half [15]. Therefore, these results highlight the dynamic aspects of goalball gameplay and offer insights into potential strategies for games and training.

The findings of the present study appear to be in line with previous scientific evidence from players without disabilities, though the limited research on DSL in players with disabilities prevents us from making direct comparisons. DSL, characterized by the dynamic and fluctuating demands placed on the body during athletic activities, has been identified as a significant contributor to fatigue among players [10], similarly to the potential contribution of this variable to the reduction in the number and intensity of impacts observed in the present study. This phenomenon encompasses various factors such as the intensity, duration, and frequency of movement patterns, as well as environmental conditions and psychological stressors [10]. Performance factors such as anthropometry, core stability, acoustic reaction time, neuromuscular control, and proprioception seem to influence the defensive roles of players, potentially affecting both impact and the workload it induces [19,20,21]. Moghadam et al. [5] demonstrated the positive effect of core stability training on goalball players’ motor performance, suggesting it should be incorporated to enhance performance and prevent injury. Improved core strength enhances both throwing and balance, aiding players in withstanding impacts during the defensive phase. This supports the link between core strength, proprioception, and workload during ground impacts. While Bataller-Cervero et al. [5,19] did not find correlations between core stability and performance, this could be because they focused on attacking phases, specifically on throwing, and not as much on defensive movements, particularly on impacts and fatigue caused by them. Therefore, it would be interesting to study the relationship between core strength and fatigue caused by impacts.

Previous evidence has reported significant reductions in goalball performance as a consequence of workload and fatigue and has shown that it may be responsible for impairing performance in key technical actions during the game [7,22]. Even perceived fatigue, not just actual fatigue, is associated with a reduced frequency of throws and a lower intensity of actions [7]. Thus, fatigue may be a factor to consider if performance is to be optimized during games. In our pilot study, we cannot assert that the high number and intensity of impacts directly cause fatigue, as fatigue was not measured. Nevertheless, the results do show that both the number and intensity of impacts decrease, possibly due to the accumulated workload caused by these impacts over the course of the game.

However, it is worth noting that, to date, no research has investigated the game factors that increase workload, nor whether it develops more during the defensive or offensive phase. An integrated and collaborative approach to workload management is necessary in team sports where players experience impacts, emphasizing the importance of ongoing research and effective monitoring to protect players’ health and optimize their performance [23,24]. This is why the present study represents a significant contribution to the field of Paralympic sports, and more specifically to goalball. Although this study has only six players, it is noteworthy that these six players represent the total number of players at the regional level competing in the highest national league.

However, several limitations should be acknowledged. The small sample size and cross-sectional design limit the generalizability of findings and preclude the establishment of causal relationships. Regarding the generalizability of the results and controlling for Type II error, we acknowledge the limitations posed by the small sample size. To achieve sufficient statistical power (1 − β > 0.80) for detecting large effect sizes, a sample of at least 35 participants would be needed. As this is a pilot study, we encourage further research with larger samples to build on our findings. With such a limited number of participants, the results may not accurately reflect the broader population of goalball players, and variations in individual performance and impact dynamics might not be adequately captured. Additionally, factors such as environmental conditions, including temperature and playing surface, as well as other variables like the level of the opponent or the outcome of the game, which can influence the number and intensity of impacts received by players, were not accounted for in this study. Future research incorporating larger sample sizes and longitudinal designs could provide more robust evidence and elucidate the temporal dynamics of trunk impacts and workload accumulation over multiple games and seasons. Therefore, monitoring several games and players with different outcomes will provide insights into how game results affect impacts and workload. On the other hand, the study’s utilization of inertial sensors represents a methodological strength, allowing for objective measurement and analysis of the dynamics of trunk impacts and movement dynamics during games.

Moreover, while inertial sensors offer valuable insights into players’ movement patterns, they might not capture all defensive movements, such as the way in which a player falls, muscle activations, movement patterns, and joint ranges of motion. Thus, incorporating measurements of the physiological variables that could affect performance, such as heart rate, oxygen consumption, or lactate levels. Therefore, data from the inertial sensors and video analysis and/or electromyography [8], along with physiological variables, could provide a more comprehensive understanding of players’ defensive strategies and the specific mechanisms underlying dynamics of trunk impacts. Lastly, another limitation should be highlighted, as differences in the number of impacts or their intensity could be related to the result of the game (victory vs. defeat). This is why presenting the results of a single game could be considered a limitation, as it is more appropriate to monitor several games with different outcomes to understand how not only fatigue and DSL but also the results affect impacts during the game.

Building on the findings of this study, future research could explore the efficacy of targeted training interventions aimed at optimizing workload and performance in goalball players. Implementing strength and conditioning programs focusing on core stability, balance, and endurance could mitigate the impact of workload and enhance players’ resilience to high-force impacts.

## 5. Conclusions

In conclusion, this pilot study found a greater number of impacts at higher impact forces during the first half, compared to a higher percentage of impacts at a low impact level in the second half. These findings highlight the importance of understanding the interplay between trunk impacts and DSL in goalball players, especially regarding defensive aspects, and represent a new discovery in the context of goalball performance analysis, but they are based on a limited sample and should be considered preliminary. Future research should explore targeted training interventions, such as strength and conditioning programs focused on core stability, strength, balance, and endurance, to improve players’ resilience to high-impact collisions and optimize workload and performance in goalball.

## Figures and Tables

**Figure 1 sports-12-00291-f001:**
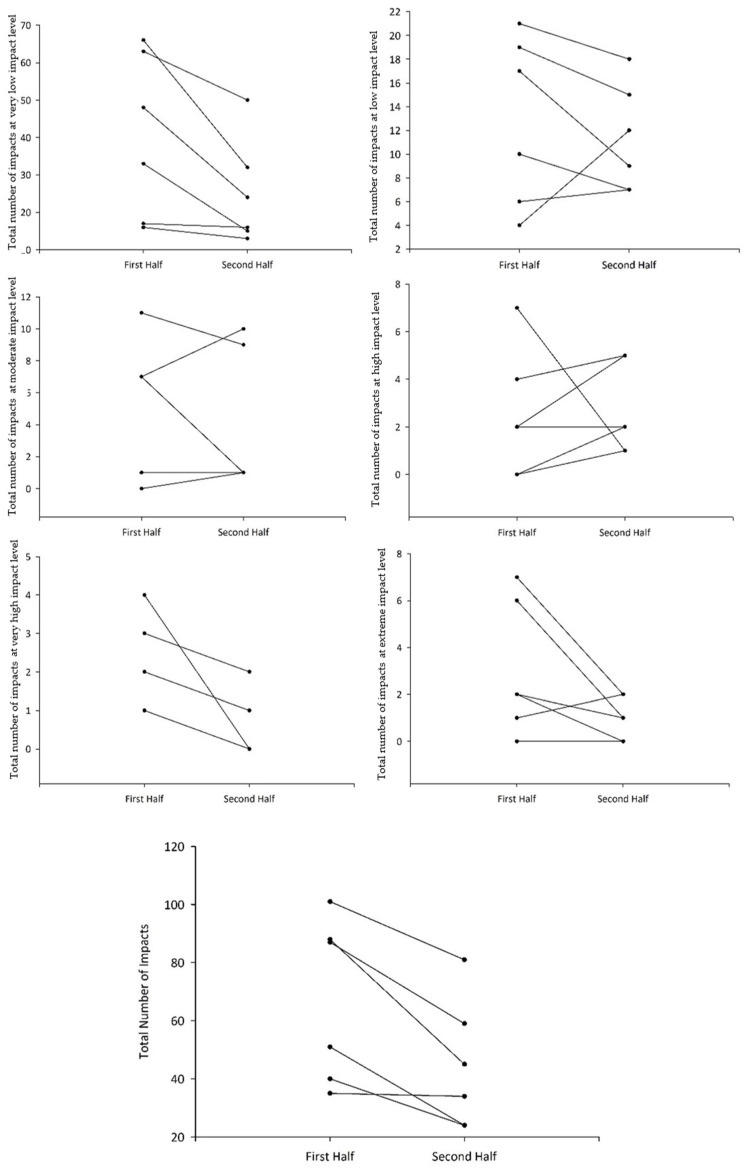
Number of impacts for each participant in the first and second halves of the goalball game.

**Table 1 sports-12-00291-t001:** Descriptive data of impact levels experienced by players and statistical comparisons between halves for impacts encountered overall and at each intensity during the game.

Impact Level Variable	Total	First Half of the Game	Second Half of the Game	*p*	ES
Total number of impacts at low impact level	104 (67.8–155)	69 (38.8–91.3)	39.5 (24–64.5)	0.028 *	0.90
Impacts at very low impact level (n°)	60 (32–101.8)	40.5 (16.8–63.8)	39.5 (24–64.5)	0.028 *	0.39
Impacts at low impact level (n°)	21.5 (15.3–35.3)	13.5 (5.5–19.5)	20 (14.5–36.5)	0.399	0.81
Impacts at moderate impact level (n°)	8 (1.8–17.8)	7 (0.8–8)	10.5 (7–15.8)	0.343	0.04
Impacts at high impact level (n°)	5.5 (1.8–8.3)	2 (0–4.8)	1 (1–9.3)	0.498	0.39
Impacts at very high impact level (n°)	3.5 (2.5–4.3)	2.5 (1.8–4)	2 (1–5)	0.023 *	0.89
Impacts at extreme impact level (n°)	3 (1.5–7.5)	2 (0.8–6.3)	0.5 (0–1.3)	0.102	0.66

Median and interquartile range include: (25th percentile–75th percentile). Significance was set at *p* < 0.05 (*) for the comparison between the first and second half of the game. ES (Effect Size): ES < 0.1 as trivial, 0.1 ≤ ES < 0.3 as small, 0.3 ≤ ES < 0.5 as medium, and ES ≥ 0.5 as large.

**Table 2 sports-12-00291-t002:** Descriptive data of the percentage of impacts in each impact levels experienced by players and statistical comparisons between halves for DSL and heart rate, and the percentage of impacts at each intensity during the game.

	First Half of the Game	Second Half of the Game	*p*	ES
Dynamic Stress Load (AU)	3.1 (2–7.2)	1.6 (1.1–3.1)	0.075	0.73
Maximum Heart Rate (beats/min)	171 (137.8–191.8)	154.5 (114.8–187.5)	0.027 *	0.90
Average Heart Rate (beats/min)	143.2 (94.2–166.5)	132.2 (100.6–159.1)	0.249	0.47
Percentage of Impacts at Very Low Impact Level	58.8 (44.3–76.9)	58 (45.5–64.7)	0.345	0.38
Percentage of Impacts at Low Impact Level	19.1 (14.1–20.7)	29.2 (19.6–31.7)	0.046 *	0.81
Percentage of Impacts at Moderate Impact Level	9.8 (0.8–15.3)	4.2 (2.7–13.1)	0.917	0.04
Percentage of Impacts at High Impact Level	3.3 (0–7.7)	5.2 (4.2–6.8)	0.344	0.39
Percentage of Impacts at Very High Impact Level	3.3 (2.3–8)	0.9 (0–3.1)	0.028 *	0.90
Percentage of Impacts at Extreme Impact Level	4 (0.8–7.9)	2.1 (0–4.6)	0.102	0.67

Median and interquartile range included (25th percentile–75th percentile). Significance was set at *p* < 0.05 (*). ES (Effect Size): ES < 0.1 as trivial, 0.1 ≤ ES < 0.3 as small, 0.3 ≤ ES < 0.5 as medium, and ES ≥ 0.5 as large.

## Data Availability

The data presented in this study are available on request from the corresponding author.

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
