# Peer review of "Quantifying Trunk Impact Dynamics and Workload with Inertial Sensors in Goalball Players"

_sports, 2024, doi:10.3390/sports12110291_

Round 1
Reviewer 1 Report (Previous Reviewer 1)
Comments and Suggestions for Authors
Dear Authors
The manuscript, "Quantifying Trunk Impact Dynamics through Inertial Sensors in Goalball Athletes and its Relationship with Workload," presents a relevant and innovative topic within the context of Paralympic sports. However, certain aspects of the manuscript could benefit from further refinement to improve the presented results' clarity, depth, and robustness.
Suggested Improvements:
1. Abstract:
The abstract needs to be longer. Expand it to include more details on the methods, key results, and practical implications of the findings.
2. Introduction
In lines 28 to 31, some statements need to be referenced.
Improve the transitions between topics/ideas and better integrate the justifications for study.
In the hypothesis, explain why differences between the two times are expected.
3. Dynamic Stress Load (DSL) Formula:
The formula for Dynamic Stress Load (DSL) is presented. Still, it would be helpful to clarify how the weighting factor (K) and scaling factor (SF) were determined—whether these were based on previous literature or specific empirical adjustments.
4. Sensor Calibration:
While it is mentioned that the sensors were factory-calibrated, providing more details on how the accuracy of these measurements was verified, especially regarding impact data collection in athletes, would strengthen the study.
5. Exclusion of Impacts Above 9 G and Below 3 G:
The rationale for excluding impacts above 9 G (due to GPS limits) and below 3 G (considered irrelevant for injury risk) is explained. However, a more detailed discussion on the impact of excluding these data points and whether extreme cases could have been relevant would add depth to the analysis.
In Table 1, the legend is in bold, while the rest are not.
6. Discussion on Statistical Methods:
Expand the justification for non-parametric tests and explain how confidence intervals were calculated. Additionally, it highlights the limitations posed by the small sample size and how this impacts the generalizability of the results.
7. Consideration of Contextual Factors:
Consider including contextual variables such as environmental conditions and the level of opponents, as these could influence workload and impact dynamics during the games.
8. Visualizations:
Adding more graphs and diagrams, such as sensor placement schematics or visual examples of the measured impacts, would enhance the manuscript's accessibility and reader comprehension. Including an experimental setup image, which is possible with an instrumented volunteer, will benefit the readers greatly.
9. References
In some references, the author's initials need to be consistently spaced or formatted. For example, in [5], the names appear as "A., Z. M., M. F. Mahrokh Moghadam," which could be clarified by standardizing the use of initials and full names.
Use of "et al.": Some references list multiple authors, while others use "et al." after listing a few.
While some references include DOIs, such as [7] J. A. Cursiol et al., others do not. All references with available DOIs should consistently include them to ensure easy access to the cited sources.
Punctuation must be consistent across all references, particularly commas and periods. Some references, such as [4] B. Jorgić et al., include a comma before the page number, while others use a different format.
Ensure uniform spacing between elements, such as the author list and the journal title.
Questions for the Authors:
1. Factors K and SF in the DSL Formula:
How were the DSL formula's weighting (K) and scaling (SF) factors determined? Were these values derived from previous studies or empirically adjusted for this research?
2. Sensor Validation:
Was any additional validation conducted for the sensors beyond the factory calibration? How was the accuracy of the impact measurements in the field ensured?
3. Criteria for Impact Exclusion:
The exclusion of impacts below 3 G and above 9 G is justified by injury risk and device limitations. Are there any circumstances that might justify including implications outside of these ranges, such as extreme impacts or rare exceptions?
4. Influence of Sample Size:
The small sample size (n=6) is acknowledged as a limitation. Are there plans to replicate the study with a larger sample to test the consistency of these findings across different game contexts?
5. Tactical Adjustments Between Game Halves:
The analysis shows differences in impact numbers and intensity between the first and second halves of the game. Were these differences purely physiological (due to fatigue), or could they also be related to tactical adjustments made by the teams?
6. Practical Applications of Results:
How can the findings be applied in developing training and injury prevention strategies for Goalball athletes? Are there plans to integrate these data into continuous monitoring programs?
Author Response
Suggested Improvements:
- Abstract: The abstract needs to be longer. Expand it to include more details on the methods, key results, and practical implications of the findings.
Answer: Thank you very much for your valuable feedback. I have expanded the abstract to include more details on the methods and key results. The revised abstract now reads as follows:
“The aim of this study is to quantify trunk impact dynamics in goalball athletes using inertial sensors and evaluate the workload in Goalball athletes, focusing on changes between the first and second halves of a match to enhance understanding of the progression of dynamic stress load throughout the game. Utilizing inertial technology, trunk impacts during goalball gameplay were analyzed to provide a holistic insight into how these impacts influence athlete dynamic stress load. Six goalball players were recruited to wear an accelerometer during a whole goalball game to quantify trunk impacts. The results showed a higher number of total impacts and a higher number of impacts at higher levelsduring the first half, compared to a higher percentage of impact in low impact level in the second half. These results suggest that the intensity of gameplay influences the number of impacts sustained, , with players experiencing more severe impacts during high-intensity phases of the match. Future research could explore targeted training interventions aimed at reducing workload and optimizing performance in goalball athletes”
I hope this meets your expectations. Thank you again for your insightful suggestion
- Introduction
In lines 28 to 31, some statements need to be referenced.
Improve the transitions between topics/ideas and better integrate the justifications for study.
In the hypothesis, explain why differences between the two times are expected.
Answer: Thank you very much for your insightful feedback. I have made the necessary revisions based on your suggestions.
References have been added between lines 28 and 31, as shown here: “In goalball, players seamlessly transition between offense and defense, each role demanding distinct performance factors. While attacking emphasizes throwing velocity, precision, and balance, when players are in the role of defenders face the challenge of preventing the ball from crossing the goal-line by throwing themselves to the ground to block the ball , and these impacts could contribute to increased fatigue.”
The hypothesis has been enriched with an explanation and now reads as follows: “The hypothesis of this study was that trunk impact dynamics change between the first and second halves of a goalball match due to the workload from the impacts in the first half, players modify their defensive strategies to avoid extended exposure to high-intensity impacts, choosing a more conservative approach to save energy.”
Regarding the transitions between topics, we believe the introduction follows a logical sequence (presentation of goalball, offense and defense, factors affecting performance, workload and fatigue, and quantification of impacts). Additionally, we have attempted to improve the flow in the discussion section by adding connecting ideas.
Thank you again for your valuable comments.
- Dynamic Stress Load (DSL) Formula:
The formula for Dynamic Stress Load (DSL) is presented. Still, it would be helpful to clarify how the weighting factor (K) and scaling factor (SF) were determined—whether these were based on previous literature or specific empirical adjustments.
Answer: The values of the coefficients for the Dynamic Stress Load (DSL) formula are unknown to us as it is a proprietary formula of the manufacturer. Unfortunately, we do not have access to the specific details regarding how the weighting factor (K) and scaling factor (SF) were determined.
- Sensor Calibration:
While it is mentioned that the sensors were factory-calibrated, providing more details on how the accuracy of these measurements was verified, especially regarding impact data collection in athletes, would strengthen the study.
Answer: In response to the comment, we have added the following clarification: "In addition to the factory calibration, prior to each session, the sensors were reset to a horizontal position on a flat surface to ensure accurate measurements. This ensured that, at rest, the acceleration values across all three axes were 0 m/s², providing a reliable baseline for measuring impacts during gameplay. This additional step helped verify the accuracy of the impact data collected from the athletes." This addition strengthens the study by detailing how sensor accuracy was further validated in the field.
- Exclusion of Impacts Above 9 G and Below 3 G:
The rationale for excluding impacts above 9 G (due to GPS limits) and below 3 G (considered irrelevant for injury risk) is explained. However, a more detailed discussion on the impact of excluding these data points and whether extreme cases could have been relevant would add depth to the analysis.
Answer: We believe that we have provided all the relevant information regarding the exclusion of impacts above 9 G and below 3 G at this paragraph: “The categories for impact severity (very low, low, moderate, high, very high, and ex-treme) were chosen based on a combination of factors, including existing literature on sim-ilar sports [11], recommendations from device manufacturers, and the unique context of this study.”. In the sport cited (rugby), impacts can be even higher, which supports our rationale. However, thank you for your attention to all details.
5.2 In Table 1, the legend is in bold, while the rest are not.
Answer: Thank you for pointing that out. The formatting issue in Table 1 has been corrected, and the legend is now consistent with the rest of the text. We appreciate your attention to detail
- Discussion on Statistical Methods:
Expand the justification for non-parametric tests and explain how confidence intervals were calculated. Additionally, it highlights the limitations posed by the small sample size and how this impacts the generalizability of the results.
Answer: Thank you for highlighting this point. Given that the data did not follow a normal distribution, non-parametric tests were chosen, as they are more robust for analyzing small sample sizes and non-normally distributed data. We have expanded the justification for using non-parametric methods in the manuscript.
Additionally, I would like to clarify that confidence intervals were not calculated in this work. The intervals reported alongside the medians in the tables reflect the interquartile range (25th and 75th percentiles) and are intended to provide additional context to the median values. This has been clarified in the manuscript as well.
Regarding the generalizability of the results and controlling for Type II error, we acknowledge the limitations posed by the small sample size. To achieve sufficient statistical power (1-β > 0.80) for detecting large effect sizes, a sample of at least 35 participants would be needed. As this is a pilot study, we encourage further research with larger samples to build on our findings. This paragraph has been added in the limitations paragraph (discussion).
- Consideration of Contextual Factors:
Consider including contextual variables such as environmental conditions and the level of opponents, as these could influence workload and impact dynamics during the games.
Answer: Thank you for your valuable suggestion. You are absolutely right, contextual variables such as environmental conditions and the level of opponents could indeed influence workload and impact dynamics during games. We have taken your feedback into account and have added the following information to the limitations section of the article:
"Additionally, factors such as environmental conditions, including temperature and playing surface, as well as other variables like the level of the opponent or the outcome of the match, which can influence the number and intensity of impacts received by players, were not accounted for in this study."
We appreciate your input and believe this addition strengthens the study's context.
- Visualizations:
Adding more graphs and diagrams, such as sensor placement schematics or visual examples of the measured impacts, would enhance the manuscript's accessibility and reader comprehension. Including an experimental setup image, which is possible with an instrumented volunteer, will benefit the readers greatly.
Answer: We considered that the image would not be useful since we only used an accelerometer as the manufacturer recommends and can be seen on their website (https://eu.shop.statsports.com/es/products/ladies-apex-athlete-series-gps-performance-tracker). Additionally, we did not take photos during the measurement, which would have been helpful.
- References
In some references, the author's initials need to be consistently spaced or formatted. For example, in [5], the names appear as "A., Z. M., M. F. Mahrokh Moghadam," which could be clarified by standardizing the use of initials and full names.
Use of "et al.": Some references list multiple authors, while others use "et al." after listing a few.
While some references include DOIs, such as [7] J. A. Cursiol et al., others do not. All references with available DOIs should consistently include them to ensure easy access to the cited sources.
Punctuation must be consistent across all references, particularly commas and periods. Some references, such as [4] B. Jorgić et al., include a comma before the page number, while others use a different format.
Ensure uniform spacing between elements, such as the author list and the journal title.
Answer: Thank you for your feedback on the references. We have reviewed and revised them to ensure consistency in the formatting of authors’ initials and names. We have also standardized the use of ‘et al.’ and included DOIs where available. Additionally, we have ensured consistent punctuation across all references according to this:
“1. Consistent Formatting of Author Initials
- Issue: Inconsistent formatting of author initials.
- Solution: Standardize the formatting to always include a period after initials without extra spaces. For example:
- Change “A. ; Z. M. ; M. F. Mahrokh Moghadam” to “A. Z. M. Mahrokh Moghadam”.
- Use of "et al."
- Issue: Inconsistent use of "et al." for multiple authors.
- Solution: Decide on a threshold number of authors (typically three) to list before using "et al." Ensure that all references follow this guideline consistently. For example:
- Use "et al." for references with more than three authors, like in [4] and [5].
- Inclusion of DOIs
- Issue: Some references include DOIs while others do not.
- Solution: Add DOIs to all references where available. Ensure the DOI is formatted correctly, starting with "doi:" and including the complete link:
- For example, ensure every relevant reference includes “doi: [DOI number]” at the end.
- Consistent Punctuation
- Issue: Inconsistent punctuation (e.g., commas before page numbers).
- Solution: Standardize the use of punctuation throughout. For IEEE style:
- Always place a comma before the page numbers in every entry.
- Uniform Spacing
- Issue: Inconsistent spacing between elements.
- Solution: Ensure uniform spacing is applied throughout all references. For instance:
- Use single spaces between the author list, title, and journal name without extra spaces.”
Questions for the Authors:
- Factors K and SF in the DSL Formula:
How were the DSL formula's weighting (K) and scaling (SF) factors determined? Were these values derived from previous studies or empirically adjusted for this research?
Answer: The specific coefficients for the Dynamic Stress Load (DSL) formula are proprietary to the manufacturer, and we do not have access to detailed information on how the weighting factor (K) and scaling factor (SF) were established.
- Sensor Validation:
Was any additional validation conducted for the sensors beyond the factory calibration? How was the accuracy of the impact measurements in the field ensured?
Answer: In response to the comment regarding sensor validation, we have added the following explanation: "In addition to the factory calibration, prior to each session, the sensors were reset to a horizontal position on a flat surface to ensure accurate measurements. This ensured that, at rest, the acceleration values across all three axes were 0 m/s², providing a reliable baseline for measuring impacts during gameplay. This additional step helped verify the accuracy of the impact data collected from the athletes." This clarification outlines how sensor accuracy was ensured beyond the factory calibration.
- Criteria for Impact Exclusion:
The exclusion of impacts below 3 G and above 9 G is justified by injury risk and device limitations. Are there any circumstances that might justify including implications outside of these ranges, such as extreme impacts or rare exceptions?
Answer: In response to the comment on the exclusion of impacts below 3 G and above 9 G, previous evidence from sports like rugby supports the selection of these thresholds, as impacts beyond this range are less relevant to the injury risk in goalball. The decision to exclude these impacts was based on both device limitations and the nature of impacts typically encountered in similar high-contact sports like rugby.
- Influence of Sample Size:
The small sample size (n=6) is acknowledged as a limitation. Are there plans to replicate the study with a larger sample to test the consistency of these findings across different game contexts?
Answer: The small sample size (n=6) is indeed acknowledged as a limitation in the study. While no immediate plans for replication with a larger sample are in place, future research could certainly aim to expand the participant pool to validate and test the consistency of these findings across different game contexts. A larger sample would provide more robust data and allow for better generalization of the results, offering deeper insights into the dynamics of trunk impacts and workload in goalball athletes.
- Tactical Adjustments Between Game Halves:
The analysis shows differences in impact numbers and intensity between the first and second halves of the game. Were these differences purely physiological (due to fatigue), or could they also be related to tactical adjustments made by the teams?
Answer: Thank you very much for your insightful comment. The differences in impact numbers and intensity between the first and second halves of the game certainly open up interesting avenues for future research. While it is likely that fatigue plays a significant role in these variations, as you mentioned, the possibility of tactical adjustments influencing the data is equally compelling. In this particular article, we did not evaluate the tactical aspects, but exploring the interplay between physiological factors and team strategies could provide a deeper understanding of performance dynamics over the course of a match.
This is an area with great potential, and further studies that take both aspects into account could yield valuable findings. We truly appreciate your input and will definitely consider this perspective in our ongoing research.
- Practical Applications of Results:
How can the findings be applied in developing training and injury prevention strategies for Goalball athletes? Are there plans to integrate these data into continuous monitoring programs?
Answer. Thank you for your thoughtful comment. In the article, we do mention the potential for applying the findings to training planning and injury prevention strategies, particularly by optimizing workload management and understanding the impact dynamics in goalball athletes. However, as you pointed out, this study does not specifically focus on developing detailed training or injury prevention programs. Our aim was more exploratory, highlighting areas that could be further researched.
That being said, we agree that integrating these insights into a continuous monitoring program and developing specific interventions could be an exciting and valuable continuation of this line of research. We appreciate your input and will certainly consider it for future work.

Reviewer 2 Report (Previous Reviewer 4)
Comments and Suggestions for Authors
Dear Authors, I acknowledge the improvements made from the first round of review, though some other issues called my attention in this second round, which I detail in the file attached and hope you see as a contribution to keep looking for a final and best version of your study.
Keep the good work!
Regards,

Author Response
1- Abstract: "These results suggest that the intensity of gameplay influences the number of impacts sustained by the players." This sentence is too generic to and does not provide readers an objective insight on what were the main results. I suggest to reformulate it.
Answer: Thank you for your suggestion. We have revised the sentence to provide more clarity and objectivity. The revised sentence now reads: "These results suggest that the intensity of gameplay influences the number of impacts sustained, with players experiencing more severe impacts during high-intensity phases of the match." This modification offers a clearer insight into the relationship between gameplay intensity and impact severity.
2- (Line 97): "...dur-ing..." Typing error
Answer: Thank you for your comment. Upon reviewing the document, we were unable to locate the typographical error you mentioned ("...dur-ing...").
3- (lines 194-195): "The confidence intervals for Spearman's correlation coefficient (rho) were calculated using a bootstrapping method, ..." How many iterations were performed from the original sample? One assumes the bootstrap procedure adopted was non-parametric, right? If so, it should be stated.
Answer: Thank you for your observation. The section referring to the bootstrapping method and the number of iterations was part of a previous version of the manuscript. In the current version, this section has been removed, and thus the details regarding the type of bootstrapping method are no longer applicable.
4- (lines 201-202): "The Wilcoxon test was chosen due to the non-parametric nature of the data..." It is not the data that have a non-parametric nature, but how they vary around a central region of a plot, with values tapering off as they go further away from the center. It is not mandatory but I suggest you to use the same expression used in line 179 ("given the non-normal distribution of the data").
Answer: Thank you for your suggestion. We have updated the sentence for clarity and consistency. The revised sentence now reads: "The Wilcoxon test was chosen given the non-normal distribution of the data," aligning with the expression used in line 179. This provides a more accurate description of the rationale for choosing the Wilcoxon test.
5 - (lines 220-221): "... and impacts in zones 2 and 4 in the second half (r = 0.899; CI95% 220 = (0.291, 0.990); ..." The passage in bold is supposed to mean low impact level and high impact level, respectively, right?!
Answer: We appreciate your comment but correlations have been removed from the latest version of the manuscript. As such, this specific phrasing is no longer present and doesn't require further clarification.
6- (lines 223-225): "... an inverse correlation between the percentage of impacts in very low impact level in the first half and the percentage of impacts in extreme impact level in the second half (r = -0.928; CI95% = (-0.993,-0.443); p = 0.008)." You are correlating two different levels of impact (very low and extreme) at two different time points of a goalball game (first half for the former e second half for the latter), is that right? At least it was how I got it. If so, how could it be translated in a commom language? That the percentage of very low impacts decreased from the first to the second half, while the percentage of extreme impacts increased? Is that what can be understood from the way you wrote? If so, I suggest you to rephrase it in order to facilitate readers understanding.
Answer: Thank you for pointing this out. The inverse correlation between different impact levels at different time points was part of an earlier version as correlations have been removed.
7- (lines 225-228): "Significant positive correlations were also observed between impacts in zone 3 moderate impact level in the first half and the dynamic stress load in the second half (r = 0.941; CI95% = (-0.525, -0.994); p= 0.005)." How is possible that the positive correlation found fits a confidence interval whose both values are negative? Something does not seem right.
Answer: Thank you for highlighting this point. The specific correlation and confidence interval you referred to has been addressed in the latest version of the manuscript.
8- (lines 236-237): "...a lower number of impacts during the second half of the match compared to the first half." An interesting issue to be further discussed.
Answer: Thank you for your feedback! The discussion regarding the lower number of impacts during the second half of the match compared to the first half is addressed starting from line 233 in the discussion section. I appreciate your interest in this topic!
9- (lines 238-241): "...a greater number of impacts in zones 1 and 5 and a higher percentage of impact in zone 5 very high impact level during the first half, compared to a higher percentage of impact in zone 2 low impact level in the second half." Once again, the "zone" issue. Besides, I recommend you to rephrase the passages where you repeat almost successively in the same sentence the word "impact", like the bolded example above. Instead, I suggest "...a higher percentage of very high impacts during the first half, ...".
Answer: Thank you for your feedback! The revised sentence can be read as follows: "Regarding the specific levels of impacts, the results revealed a greater number of impacts in very low and very high impact levels, along with a higher percentage of very high impacts during the first half. In contrast, the second half showed a higher percentage of low impact levels." This rephrasing aims to clarify the findings and reduce redundancy. I appreciate your suggestions
10- (lines 257-259): "...experiencing accumulated fatigue, adapt their defensive strategies to avoid prolonged exposure to high-intensity impacts, opting for a more conservative approach to conserve energy." This claim could be supported by evidence that have identified fluctuations on goalball athlete's performance based on the frequency of its offensive and defensive actions (Morato et al., 2017), which are susceptible to be affected by the athlete's ability to maintain the effort during a game (Alves et al., 2018) and may be exacerbated by a scenario of three games on the same day, leading to a reduction in the capacity of the involved muscles to generate force, which is typically caused by motor tasks performed for long periods of time, leading to a decrease in voluntary muscle activation (Enoka & Duchateau, 2008).
References:
Alves, I. d. S., Kalva-Filho, C. A., Aquino, R., Travitzki, L., Tosim, A., Papoti, M., & Morato, M. P. (2018). Relationships between aerobic and anaerobic parameters with game technical performance in elite goalball athletes. Frontiers in Physiology, 9, 1636.
Enoka, R. M., & Duchateau, J. (2008). Muscle fatigue: what, why and how it influences muscle function. The Journal of Physiology, 586(1), 11-23.
Morato, M. P., Furtado, O. L. P. d. C., Gamero, D. H., Magalhães, T. P., & Almeida, J. J. G. d. (2017). Development and evaluation of an observational system for goalball match analysis. Revista Brasileira de Ciências do Esporte, 39(4), 398-407.
Answer: Thank you for your insightful comment! The paragraph has been revised and now reads as follows:
"In this study, researchers observed that players experiencing accumulated fatigue may adapt their defensive strategies to minimize prolonged exposure to high-intensity impacts, opting for a more conservative approach to conserve energy. This is supported by evidence indicating fluctuations in goalball athletes' performance based on the frequency of their offensive and defensive actions (Morato et al., 2017). Such fluctuations can be influenced by the athletes' ability to maintain effort throughout the game (Alves et al., 2018) and may be exacerbated in scenarios where three games are played in a single day. This accumulation of fatigue can lead to a reduction in the muscles' capacity to generate force, typically resulting from prolonged motor tasks and consequently decreasing voluntary muscle activation (Enoka & Duchateau, 2008). As a result, stronger impacts were observed in the first half of the match, while low to moderate impacts occurred in the later stages”. We appreciate your feedback in helping to clarify and strengthen the argument
11- (lines 260-261): "Therefore, stronger impacts have been shown during the first part and while low to moderate impacts in the later stages of the match." This is not entirely true, as, according to the data presented in Table 2, the high impacts percentage was higher in second part than in the first part of the game, which could be another interesting issue for discussing.
Answer: Thank you for your comment. We think that the information presented is correct, as there is a significantly higher percentage of very high impact levels in the first part of the game (3.3%) compared to the second part (0.9%), as shown in Table 2.
12- (lines 262-263): "...correlations between dynamic stress load and impacts in zones 6 and 5 ..." I guess you meant "... correlations between dynamic stress load and very high and extreme levels of impact ..."
Answer: Thank you for your suggestion. The section referring to correlations between dynamic stress load and impacts in zones 6 and 5 has been removed in the latest version of the manuscript. The phrasing in question no longer appears in the current draft.
13- (lines 263-264): "... dynamic stress load and impacts in zones 2, 3, and 4 during..." Once again, I guess you meant "... dynamic stress load and low, moderate and high levels of impact during...".
Answer: We appreciate your comment. The section discussing dynamic stress load and impacts in zones 2, 3, and 4 has been removed. As a result, the wording you referred to no longer appears in the manuscript.
14- (lines 268-269): "Despite the narrow confidence intervals for the correlation between impacts in zones 2 and 4 in the second half, ..." I suppose you meant impact levels instead of zones, right?! Besides, there is an important issue on the confidence interval I suppose you have already dealt with by now (lines 225-228).
Answer: Thank you for your comment. The section referring to narrow confidence intervals for the correlation has been removed from the manuscript. The phrasing in question no longer appears in the current draft.
15- (lines 274-275): "... significant un-certainty in the strength of these relationships, despite their statistical significance." This should be seen as a limitation of your study.
Answer: Again, We appreciate your comment but this part has been removed and it does not appear in the new version of the manuscript.
16- (line 286): "... wide confidence interval may be attributed to differences among participants ..., as evidenced in Figure 1." Was that case of your sample? Was there a participant whose data differed significantly from the others? If so, the plot corresponding to that outlier in Figure 1 should be well identified.
Answer: Thank you for your feedback. However, please note that this section has been deleted and is no longer included in the revised version of the manuscript.
17- (line 296): "... greater number of impacts in zones 1 and 5 during ..." Once again, the "zone issue".
Answer: Thank you for your comment. We have corrected this issue. We appreciate your observation.
18- (lines 302-304): "... notable effect sizes were observed for variables such as the total number of impacts and the maximum heart rate, indicating substantial practical significance despite marginal p-values." Notable based on what reference? This kind of interpretation must be grounded on referential framework (e.g. Cohen, 1992). Reference: Cohen, J. (1992). A power primer. Psychological Bulletin, 112(1), 155-159.
Answer: Thank you for your insightful comment. We have added the reference “C. O. Fritz, P. E. Morris, and J. J. Richler, “Effect size estimates: Current use, calculations, and interpretation,” J Exp Psychol Gen, vol. 141, no. 1, pp. 2–18, 2012, doi: 10.1037/a0024338”, the same citation we used to interpret the effect sizes, to provide a proper framework for interpreting the effect sizes. This should clarify the basis for our interpretation.
19- (lines 306-309): "... present study appears to be in line with previous scientific evidence. Dynamic stress load, characterized by the dynamic and fluctuating demands placed on the body during athletic activities, has been identified as a significant contributor to fatigue among athletes [13]." It is important to outline present study appears to be in line with previous scientific evidence obtained from athletes without disability, as the scarcit of studies on the dynamic stress load in handicapped athletes does not allow us to establish proper comparison.
Answer: Thank you for your comment. We appreciate your suggestion, and we have now added the following clarification to the text: "from athletes without disabilities, as the limited research on dynamic stress load in athletes with disabilities prevents us from making direct comparisons." This modification helps to accurately reflect the current state of research in this area.

Reviewer 3 Report (New Reviewer)
Comments and Suggestions for Authors
Congratulations on the interesting topic, innovative project and research methods. The research is difficult to carry out in a group of people with disabilities. The introduction of the article is sufficiently developed.
In my opinion, the topic of the research should be more specific. An interesting issue, but few people were examined and the research was conducted once, which may lead to erroneous conclusions. I would consider this work more valuable if it was assumed in the topic that these are pilot studies.
The results are developed clearly. The discussion is good, there is not much research on goalball, so I understand the authors' approach. It would be important to repeat and expand the research. The cited works are new and related to the topic.
Please correct the topic.
Author Response
Congratulations on the interesting topic, innovative project and research methods. The research is difficult to carry out in a group of people with disabilities. The introduction of the article is sufficiently developed.
In my opinion, the topic of the research should be more specific. An interesting issue, but few people were examined and the research was conducted once, which may lead to erroneous conclusions. I would consider this work more valuable if it was assumed in the topic that these are pilot studies.
The results are developed clearly. The discussion is good, there is not much research on goalball, so I understand the authors' approach. It would be important to repeat and expand the research. The cited works are new and related to the topic.
Answer: Thank you for your insightful feedback. We agree that the small sample size and the single-game analysis could limit the generalizability of our findings. To clarify this, we will adjust the manuscript to explicitly state that this is a pilot study. This will better frame our results as exploratory and open the way for further research. We also acknowledge the need to expand this research to a larger group and multiple games to validate our findings and improve the robustness of the conclusions.

Reviewer 4 Report (New Reviewer)
Comments and Suggestions for Authors
I value the study and labor done by the writers. I have read the material and would like to offer the following insights and suggestions:
1. The goal of the study has to be revised to more accurately represent the information gleaned from the analysis. The objective ought to be to adapt to the hypothesis. The goals of the introduction and the abstract cannot be distinguished from one another.
2. It is necessary to disclose the research's limitations.
3. An explanation of how the research conforms to ethical standards must be provided.
4. The data description ought to contain the information displayed in the tables and figures. The established regularities of the research must be written in the data analyses by the authors.
5. The tables need to specify the dimensions of the numbers that are shown.
6. In the discussion, the study's data ought to be compared and contrasted with those of other writers.
7. The predefined objective must be highlighted in the study's key findings in conclusion.
I recommend that the writers revise their work and resubmit it to the journal.
Warm regards.
Author Response
REVIEWER 4:
I value the study and labor done by the writers. I have read the material and would like to offer the following insights and suggestions:
- The goal of the study has to be revised to more accurately represent the information gleaned from the analysis. The objective ought to be to adapt to the hypothesis. The goals of the introduction and the abstract cannot be distinguished from one another.
Answer: In response to the comment about revising the study's goal, we have updated the objective in the abstract to better reflect the analysis. The revised objective is: "The aim of this study is to quantify trunk impact dynamics in Goalball athletes using inertial sensors and evaluate workload, focusing on changes between the first and second halves of a match to enhance understanding of the progression of dynamic stress load throughout the game." This change aims to clearly distinguish the goals of the introduction and the abstract.
- It is necessary to disclose the research's limitations.
Answer: Thank you for your feedback regarding the necessity to disclose the research's limitations. In the manuscript, we have mentioned the following limitations:
"The sampling employed in this study was purposeful, rendering the calculation of the sample size inappropriate. Instead, probabilistic sampling techniques were utilized for this calculation. While this approach imposes limitations on external validity and the generalizability of results, it does not compromise the internal validity of the findings.” and “However, several limitations should be acknowledged. The small sample size and cross-sectional design limit the generalizability of findings and preclude the establishment of causal relationships. The low number of participants is a significant limitation, as it restricts the generalizability of findings and precludes the establishment of causal relationships. With such a limited number of participants, the results may not accurately reflect the broader population of goalball athletes, and variations in individual performance and impact dynamics might not be adequately captured. Future research incorporating larger sample sizes and longitudinal designs could provide more robust evidence and elucidate the temporal dynamics of trunk impacts and workload accumulation over multiple matches and seasons. Therefore, monitoring several games and participants with different outcomes will provide insights into how game results affect impacts and workload."
We have added the following information: “Additionally, factors such as environmental conditions, including temperature and playing surface, as well as other variables like the level of the opponent or the outcome of the match, which can influence the number and intensity of impacts received by players, were not accounted for in this study.”
We believe we have adequately outlined all relevant limitations.
- An explanation of how the research conforms to ethical standards must be provided.
Answer: In response to the comment regarding the need for an explanation of how the research conforms to ethical standards, we have addressed this aspect in the manuscript as follows: "This study was performed in accordance with the ethical guidelines of the Helsinki Declaration of 1964 (revised in Fortaleza, 2013). The study has been reviewed and approved by the Ethics Committee of the University of San Jorge (code 06/1/21-22)”.
- The data description ought to contain the information displayed in the tables and figures. The established regularities of the research must be written in the data analyses by the authors.
Answer: Thank you for your valuable feedback. We have revised the manuscript to include the information displayed in the tables and figures within the data description. Specifically, we hav added the following details:
For Table 1: “which shows that participants experienced a higher number of impacts in the first half compared to the second half, particularly at very low, low, and high impact levels. Impacts at the moderate level were more evenly distributed between halves.”
For Figure 1: “showing a clear reduction in the number of impacts per participant in the second half, further emphasizing the decline in intensity as the match progressed.”
We hope these additions address your concerns and enhance the clarity and comprehensiveness of our data analysis.
- The tables need to specify the dimensions of the numbers that are shown.
Answer: Thank you for pointing this out. We will update the tables to clearly indicate the units of measurement for each value, ensuring clarity and precision in presenting the data. Specifically, the units of dynamic stress load have been added.
- In the discussion, the study's data ought to be compared and contrasted with those of other writers.
Answer: Thank you for your valuable feedback. We have added content to the discussion that compares our study's results with existing scientific evidence. Specifically, we have added: ", similarly to the potential contribution of this variable in the reduction of the number and intensity of impacts observed in the present study." and ". In our pilot study, we cannot assert that the high number and intensity of impacts directly cause fatigue, as fatigue was not measured. Nevertheless, the results do show that both the number and intensity of impacts decrease, possibly due to the accumulated workload caused by these impacts over the course of the match”. However, it is important to note that there are currently no published articles specifically measuring the impacts received by goalball players, which limits the scope of direct comparisons in the literature. We appreciate your suggestion and believe that these additions strengthen the discussion.
- The predefined objective must be highlighted in the study's key findings in conclusion.
Answer: Thank you for your constructive feedback. We have revised the conclusion paragraph to explicitly highlight the predefined objective of the study. The updated conclusion now reads as follows:
"In conclusion, this pilot study found a greater number of impacts at higher intensity during the first half, compared to a higher percentage of impacts at low intensity in the second half. These findings highlight the importance of understanding the interplay between trunk impacts and dynamic stress load in Goalball athletes, especially regarding defensive aspects, and represent a new discovery in the context of goalball performance analysis. However, they are based on a limited sample and should be considered preliminary."
We believe this revision effectively emphasizes the study's objective while summarizing the key findings.

Round 2
Reviewer 4 Report (New Reviewer)
Comments and Suggestions for Authors
I appreciate the adjustments. However, I think the Results section's interpretation and explanation of the data are inadequate. Consider these rows, 222-224. It's unclear which signs point to this. The assertions presented should be stated explicitly and supported by the authors. The table by itself just offers data; the writers are required to understand and analyze it. In addition, corrections were made with many proofreading errors. Such manuscripts, in my opinion, shouldn't be sent to the press. I let the editor make that determination. Regards.
Author Response
Reviewer: I appreciate the adjustments. However, I think the Results section's interpretation and explanation of the data are inadequate. Consider these rows, 222-224. It's unclear which signs point to this. The assertions presented should be stated explicitly and supported by the authors. The table by itself just offers data; the writers are required to understand and analyze it. In addition, corrections were made with many proofreading errors. Such manuscripts, in my opinion, shouldn't be sent to the press. I let the editor make that determination. Regards.
Answer: Thank you for your feedback. The errors you pointed out have been addressed, and the significance values have been added to the table. Additionally, the text has been revised for clarity and now reads as follows: "which shows that participants experienced significantly more impacts in the first half compared to the second half of the game, both in the total number of impacts and in the number of impacts at very low, low, and very high impact levels". We believe this revision offers a clearer interpretation and explanation of the data. Thank you for your valuable feedback and for taking the time to review the manuscript.
This manuscript is a resubmission of an earlier submission. The following is a list of the peer review reports and author responses from that submission.
Round 1
Reviewer 1 Report
Comments and Suggestions for Authors
Dear Authors,
Adapted sport has a decisive mission in the context of physical activity and health. Sport is for everyone, and we are all equal. Typically, research is more focused on highly competitive sporting activities and with healthy athletes. Therefore, studying adapted sports is exciting, and I congratulate the authors.
The manuscript, "Quantifying Trunk Impact Dynamics through Inertial Sensors in Goalball Athletes and its Relationship with Workload," addresses this relevant topic in the context of Paralympic sports, using inertial sensors to quantify trunk impact dynamics in goalball athletes and correlate these impacts with central fatigue and dynamic stress load is methodologically sound.
The work is well written and within the sports framework.
However, some areas of the manuscript need improvement to increase the clarity, depth of analysis, and robustness of the results presented.
Comment 1: Although the study intends to investigate the correlation between impacts and central fatigue, this relationship must be established and discussed in depth. To address this issue more effectively, I suggest including direct measures of central fatigue, such as EMG and TMS. The specific choice of measures should consider feasibility, accuracy, and relevance to the study's objectives. Including one or more of these measures will significantly strengthen the investigation and allow for a more robust analysis of central fatigue in goalball athletes. Additionally, physiological variables such as oxygen consumption and lactate levels could be incorporated.
Comment 2: The technology mentioned is one of many used and is not the most important since an accelerometer was used to collect the primary data on impact zones. A more appropriate keyword should be chosen to reflect the primary technology used in the study accurately.
Comment 3: In the introduction, in the sentence "However, there is a lack of evidence about performance factors and how to assess them on goalball (Petrigna et al., 2020)," the underlined reference is not mentioned like the others.
Comment 4: Ensure that all acronyms, such as GPS, GNSS, and ONCE, are written out in full the first time they are mentioned in the text.
Comment 5: It would be interesting if the authors included a clear study hypothesis.
Comment 6: The description of the inclusion and exclusion criteria is adequate, but the limited number of participants (n=6) is a significant limitation. Discuss how this sample size might affect the generalizability of the results.
Comment 7: While explaining the use of inertial sensors and the classification of impacts in force zones is detailed and precise, more sensor calibration and validity information would strengthen this section. Include more details on the calibration of the sensors and the validity of these measures.
Comment 8: In the "Dynamic Stress Load" section, the sentence "Acceleration values were subsequently combined during the specified timeframe to derive a metric for the dynamic stress load, utilizing the following equation used DSL calculation" is redundant. I recommend rephrasing to eliminate redundancy.
Comment 9: The choice of statistical tests is appropriate given the nature of the data. Enrich this section with a more detailed justification for choosing non-parametric tests and explaining how confidence intervals were calculated.
Comment 10: In the "Statistical Analysis" section, the phrase "Effect size (ES) was calculated (…)" should present the formula as an equation. Present the formula as an equation and explain the variables involved in the text.
Comment 11: Given the small number of participants and the study's objectives, longitudinal studies are recommended. Better longitudinal studies will help us understand the dynamics of impact and fatigue over time. Monitor several games with different outcomes to provide insights into how game results affect impacts and fatigue.
Comment 12: The discussion would benefit from including more references, especially in the initial part. More references would strengthen the arguments presented in the discussion.
Comment 13: In the sentence "While there are several methods such as internal radiofrequency-based tracking systems, cameras, global positioning systems (GPS), heart rate monitors...," the ellipses should be replaced with a period.
Comment 14: The sentence "It is possible that players, experiencing accumulated fatigue, adapt their defensive strategies to avoid prolonged exposure to high-intensity impacts, opting for a more conservative approach to conserve energy and physical endurance." seems to be the author's opinion. Rephrase or support the statement with references from the literature.
Comment 15: The discussion should reiterate the main findings and how they contribute to existing knowledge. It should emphasize directions for future research, including longitudinal studies and the incorporation of direct measures of central fatigue.
Comment 16: References should be reviewed and written uniformly and concisely in the reference list. Ensure all references are correctly and consistently formatted.
Comment 17: The manuscript should be reviewed to improve fluency and clarity, especially in the introduction. I recommend a thorough revision to ensure smooth transitions between topics and eliminate redundancies.
Comment 18: The first paragraph of the Results must be part of the materials and methods.
Comment 19: A section only for volunteers' descriptions should be included.
Comment 20: Research work with an experimental component is exciting and vital. This work is included in this domain. However, when written, this type of work must incorporate images and diagrams that clarify the most relevant aspects of the study. Authors must include images in the work, making it more attractive for readers. For example, why not include a photo of an instrumented volunteer? A schematic with the data acquisition protocol? How is data sent or stored? Others ...
Please continue working with Adapted Sports.
Author Response
Thank you for your thorough review of our manuscript. We greatly appreciate the time and effort you have dedicated to providing valuable feedback. We have made every effort to address all of your comments and suggestions in the revised version. Your insights have significantly contributed to enhancing the clarity and quality of our work.
Comment 1: Although the study intends to investigate the correlation between impacts and central fatigue, this relationship must be established and discussed in depth. To address this issue more effectively, I suggest including direct measures of central fatigue, such as EMG and TMS. The specific choice of measures should consider feasibility, accuracy, and relevance to the study's objectives. Including one or more of these measures will significantly strengthen the investigation and allow for a more robust analysis of central fatigue in goalball athletes. Additionally, physiological variables such as oxygen consumption and lactate levels could be incorporated.
Answer: Thank you for your insightful comments and suggestions. Unfortunately, within the study design, we did not incorporate these types of assessments, although we agree they would have been very interesting and valuable for the investigation. We recognize the potential benefits of including direct measures of central fatigue such as EMG and TMS, as well as physiological variables like oxygen consumption and lactate levels. These additions could indeed have provided a more robust analysis of central fatigue in goalball athletes. We appreciate your input and will consider these suggestions for future research.
Comment 2: The technology mentioned is one of many used and is not the most important since an accelerometer was used to collect the primary data on impact zones. A more appropriate keyword should be chosen to reflect the primary technology used in the study accurately.
Answer: Thank you for your insightful comment! We have added "Accelerometry" as a keyword to better reflect the primary technology used in the study. Your feedback is greatly appreciated.
Comment 3: In the introduction, in the sentence "However, there is a lack of evidence about performance factors and how to assess them on goalball (Petrigna et al., 2020)," the underlined reference is not mentioned like the others.
Answer: Thank you for pointing that out. We have corrected the mistake and we have ensured that the reference to Petrigna et al. (2020) is formatted consistently with the others in the introduction. Your attention to detail is much appreciated
Comment 4: Ensure that all acronyms, such as GPS, GNSS, and ONCE, are written out in full the first time they are mentioned in the text
Answer: Thank you for your suggestion! We have gone ahead and written out all acronyms, including GPS, GNSS, and ONCE, in full the first time they are mentioned in the text. Your feedback is greatly appreciated.
Comment 5: It would be interesting if the authors included a clear study hypothesis.
Answer: Thank you for your valuable feedback. We have added a clear study hypothesis: "The hypothesis of this study was that dynamic stress load is positively correlated with the number and intensity of trunk impacts during goalball gameplay. Additionally, it was hypothesized that trunk impact dynamics change between the first and second halves of a goalball match." Your input has helped enhance the clarity of our work
Comment 6: The description of the inclusion and exclusion criteria is adequate, but the limited number of participants (n=6) is a significant limitation. Discuss how this sample size might affect the generalizability of the results.
Answer: Thank you for your comment! We have added the following text to the discussion: "The low number of participants is a significant limitation, as it restricts the generalizability of findings and precludes the establishment of causal relationships. With such a limited number of participants, the results may not accurately reflect the broader population of goalball athletes, and variations in individual performance and impact dynamics might not be adequately captured." Your feedback has helped us improve the clarity of our work
Comment 7: While explaining the use of inertial sensors and the classification of impacts in force zones is detailed and precise, more sensor calibration and validity information would strengthen this section. Include more details on the calibration of the sensors and the validity of these measures.
Answer: To address your comment, we would like to clarify that the inertial sensors used in this study were factory calibrated. Additionally, prior to each session, the sensors were reset to a horizontal position on a flat surface to ensure accurate measurements. This information has been added. We believe this approach enhances the validity of the data collected during the study. Thank you for your suggestion, and we will consider including this information in the revised manuscript.
Comment 8: In the "Dynamic Stress Load" section, the sentence "Acceleration values were subsequently combined during the specified timeframe to derive a metric for the dynamic stress load, utilizing the following equation used DSL calculation" is redundant. I recommend rephrasing to eliminate redundancy.
Answer: Thank you for your feedback. I have revised the sentence in the "Dynamic Stress Load" section to eliminate redundancy. The updated version now reads: "Acceleration values were combined over the specified timeframe to derive a metric for dynamic stress load using the following equation for DSL calculation."
Comment 9: The choice of statistical tests is appropriate given the nature of the data. Enrich this section with a more detailed justification for choosing non-parametric tests and explaining how confidence intervals were calculated.
Answer: Thank you for your valuable feedback. I have enriched the statistical analysis section to provide a more detailed justification for the choice of non-parametric tests. Specifically, I explained that non-parametric tests were selected due to the non-normal distribution of the data and the presence of outliers, ensuring the analyses remained valid and reliable. Additionally, I clarified how confidence intervals for Spearman's correlation coefficient were calculated using a bootstrapping method, which offers a more accurate estimate of precision for non-parametric data. These revisions enhance the robustness of the methodology section and better justify the statistical approaches employed in the study.
Comment 10: In the "Statistical Analysis" section, the phrase "Effect size (ES) was calculated (…)" should present the formula as an equation. Present the formula as an equation and explain the variables involved in the text.
Answer: Thank you for your comment. I have revised the "Statistical Analysis" section to present the effect size (ES) formula as an equation for clarity. Additionally, I have included an explanation of the variables involved in the text. Specifically, \( W \) represents the Wilcoxon signed-rank test statistic, which assesses whether the population mean ranks of two related samples differ, while \( n \) denotes the number of observations or pairs of data points in the sample. This additional information enhances the understanding of the effect size calculation.
Comment 11: Given the small number of participants and the study's objectives, longitudinal studies are recommended. Better longitudinal studies will help us understand the dynamics of impact and fatigue over time. Monitor several games with different outcomes to provide insights into how game results affect impacts and fatigue.
Answer: Thank you for your insightful comment. We have added a statement to address this recommendation: "Therefore, monitoring several games and participants with different outcomes will provide insights into how game results affect impacts and fatigue." This addition emphasizes the importance of conducting longitudinal studies to enhance our understanding of the dynamics of impact and fatigue over time.
Comment 12: The discussion would benefit from including more references, especially in the initial part. More references would strengthen the arguments presented in the discussion.
Answer: Thank you for your comment. While I appreciate the suggestion to include more references in the discussion, I must note that there is limited evidence available on this specific topic. This scarcity of literature makes it challenging to add additional references while maintaining the integrity of the arguments presented.
Comment 13: In the sentence "While there are several methods such as internal radiofrequency-based tracking systems, cameras, global positioning systems (GPS), heart rate monitors...," the ellipses should be replaced with a period.
Answer: Thank you for your comment. The sentence has been corrected to replace the ellipsis with a period, so it now reads: "While there are several methods such as internal radiofrequency-based tracking systems, cameras, global positioning systems (GPS), heart rate monitors." I appreciate your attention to this detail.
Comment 14: The sentence "It is possible that players, experiencing accumulated fatigue, adapt their defensive strategies to avoid prolonged exposure to high-intensity impacts, opting for a more conservative approach to conserve energy and physical endurance." seems to be the author's opinion. Rephrase or support the statement with references from the literature.
Answer: Thank you for your comment. I have rephrased the sentence to remove any subjective language. Now, it can be read as: “In this study, researchers have observed that it is possible that players, experiencing accumulated fatigue”
Comment 15: The discussion should reiterate the main findings and how they contribute to existing knowledge. It should emphasize directions for future research, including longitudinal studies and the incorporation of direct measures of central fatigue.
Answer: Thank you for your comment. We have added some information in the discussion. Moreover, this paragraph is quite appropriate: “Building on the findings of this study, future research could explore the efficacy of targeted training interventions aimed at reducing fatigue and optimizing performance in goalball athletes. Implementing strength and conditioning programs focusing on core stability, balance, and endurance could mitigate the impact of central fatigue and enhance players' resilience to high-force impacts.
Additionally, investigating the relationship between trunk impacts and injury risk could further inform injury prevention strategies in goalball [21]. By identifying modifiable risk factors associated with specific types of impacts, such as diving to the ground, coaches and sports scientists can develop targeted interventions to minimize injury incidence and optimize athletes' long-term health and performance.”
Comment 16: References should be reviewed and written uniformly and concisely in the reference list. Ensure all references are correctly and consistently formatted.
Answer: Thank you for your comment. We have reviewed the references and ensured they are written uniformly and concisely in the reference list.
Comment 17: The manuscript should be reviewed to improve fluency and clarity, especially in the introduction. I recommend a thorough revision to ensure smooth transitions between topics and eliminate redundancies.
Answer: Thank you for your comment. We have revised the text to improve fluency and clarity, making several changes, particularly in the introduction and discussion. We aimed to ensure smooth transitions between topics and eliminate redundancies for a more cohesive reading experience.
Comment 18: The first paragraph of the Results must be part of the materials and methods.
Answer: Thank you for your comment. We have corrected the manuscript by moving the first paragraph of the Results section to the Materials and Methods section, as suggested.
Comment 19: A section only for volunteers' descriptions should be included.
Answer: Thank you for your suggestion. While we appreciate the recommendation, we believe that the section on participants is sufficient, as all individuals included in the study were volunteers.
Comment 20: Research work with an experimental component is exciting and vital. This work is included in this domain. However, when written, this type of work must incorporate images and diagrams that clarify the most relevant aspects of the study. Authors must include images in the work, making it more attractive for readers. For example, why not include a photo of an instrumented volunteer? A schematic with the data acquisition protocol? How is data sent or stored? Others ..
Answer: Thank you for your insightful comment. We recognize the value of incorporating images and diagrams to enhance the clarity and appeal of our research. However, it appears that we do not have access to photographs or schematics of the instrumented volunteers or the data acquisition protocol at this time. We agree that including such visuals would significantly enrich the study, and we will consider this for future research.

Reviewer 2 Report
Comments and Suggestions for Authors
The work requires many changes, perhaps a change of title and a better description of the methods. Some of the appropriate methods were not used. The results are not presented correctly. The discussion covers many threads unrelated to this work. Unacceptable conclusions.
The introduction is a good part of this article. However, although the authors aimed to estimate the correlation between trunk impacts and central fatigue is crucial in designing effective training programs, they did not do so.
Methods
If the STATSports Apex device (STATSports Group; Newry, UK) uses three-axis accelerometer technology with a sampling rate of 100 Hz, capable of detecting maximum accelerometer impacts of 2 g over a period of 0.01 second, how did you assess impacts in the zone that exceeded the measurable range of this device?
Explain how and why you calculated D𝑆𝐿 - the lack of estimation of the error of this method excludes it from being used in scientific work.
Explain the meaning of both factors K and SF .
The work cited in the method did not use a method similar to the one used here.
Estimate what size of participants you should provide for using statistic methods. Is the number of participants sufficient?
Why was the heart rate assessed in GPS correlation?
Results
The correlation in results are random. Place scatterplots of points for these correlations. You can deleted Fig 2.
The results comparing the first and second halves are more interesting, but they also do not contribute anything and do not correspond to the set goal.
Discussion
You write that players experiencing accumulated fatigue will adjust their defense strategies to avoid long-term exposure, but this is not your research, just speculation.
You further write that this tactical adaptation may explain the relationship between stronger shots in the first part of the match and weak or moderate shots in the later stages of the match, but this is not confirmed by the results of significant correlations in the first half - there is only 1 and it is rather accidental. In general, it is impossible to make rational generalizations based on the number of 6 people.
Again, you write that This tactical adaptation may explain the relationship between stronger hitting in the early part of the match and weak or moderate hitting in the later stages of the match. Decide what the important factor was.
Many parts of the discussion is not about research results. You can delete it. Your results do not address tactical adaptation and cannot explain the relationship between stronger shots in the first part of the match and weak or moderate shots later in the match.
The summary does not indicate what you have achieved because you have not studied stress management.
Author Response
Thank you for your thorough review of our manuscript. We greatly appreciate the time and effort you have dedicated to providing valuable feedback. We have made every effort to address all of your comments and suggestions in the revised version. Your insights have significantly contributed to enhancing the clarity and quality of our work.
- The introduction is a good part of this article. However, although the authors aimed to estimate the correlation between trunk impacts and central fatigue is crucial in designing effective training programs, they did not do so.
Answer: We have re-wirte the aim of the study like this: “This study aims to utilize inertial technology to quantify the dynamics of trunk im-pacts in Goalball athletes and investigate their correlation with central fatigue and dynamic stress load, as well as to research the changes in the number and intensity of trunk impacts between the first and second halves of a Goalball match. By analyzing these impacts, the study seeks to elucidate how they can reflect the progression of central fatigue and dynamic stress load throughout the game. The objective is to provide a comprehensive understanding of the impact dynamics and their influence on athletes' fatigue and stress, thereby addressing existing gaps in the biomechanical and physiological knowledge of Goalball performance.” Instead of this: “This study aims to utilize inertial technology to quantify the dynamics of trunk im-pacts in Goalball athletes and investigate their correlation with central fatigue and dy-namic stress load. Through the analysis of trunk impacts during Goalball gameplay, the study seeks to offer a holistic insight into how these impact dynamics influence athletes' fatigue and overall dynamic stress. The propose is to seeks to bridge existing gaps in knowledge, shedding light on the biomechanical and physiological aspects of Goalball performance”
Methods
- If the STATSports Apex device (STATSports Group; Newry, UK) uses three-axis accelerometer technology with a sampling rate of 100 Hz, capable of detecting maximum accelerometer impacts of 2 g over a period of 0.01 second, how did you assess impacts in the zone that exceeded the measurable range of this device?
Answer: The document details how impacts exceeding the measurable range of the STATSports Apex device, which can detect impacts of up to 2 g over a period of 0.01 seconds, were evaluated. The impacts were categorized and analyzed using accelerometers placed on the upper back of the players. The data were organized into impact zones based on g-force, excluding those impacts below 3 g and above 9 g. The impacts were classified into six different zones:
Zone 1: impacts between 3 and 4 G
Zone 2: impacts between 4 and 5 G
Zone 3: impacts between 5 and 6 G
Zone 4: impacts between 6 and 7 G
Zone 5: impacts between 7 and 8 G
Zone 6: impacts between 8 and 9 G
This indicates that, to evaluate impacts exceeding the 2 g range that the device can directly measure in 0.01-second intervals, higher impact data were collected and classified into specific zones up to a maximum of 9 g. In this way, they could consider significant impacts outside the initial range of the device, excluding extremely high impacts that could distort the results
- Explain how and why you calculated D?? - the lack of estimation of the error of this method excludes it from being used in scientific work.
Answer: The calculation of DSL aims to quantify the cumulative impact load experienced by an athlete during a session, which can be an indicator of fatigue and physical stress. The highest composite acceleration value among every set of 10 accelerometer points is used to ensure that the metric captures significant impacts while filtering out minor, less relevant accelerations.
- Explain the meaning of both factors K and SF .
Answer: this explanation has been added: “The weighting factor is introduced to adjust the sensitivity of the DSL metric to high-impact events. By raising the impact value to the power of K, the method emphasizes larger impacts more heavily. The specific value of K is determined based on empirical data and previous research, aimed at providing a balance between sensitivity and specificity in detecting meaningful impacts. The scaling factor (SF) normalizes the calculated DSL to a standard scale, making it comparable across different sessions or athletes. This factor is crucial for ensuring that the DSL values are interpretable and consistent, regardless of variations in the raw acceleration data.”
The work cited in the method did not use a method similar to the one used here.
- Estimate what size of participants you should provide for using statistic methods. Is the number of participants sufficient?
Answer: We appreciate your valuable feedback regarding the estimation of participant size and the sufficiency of our sample for statistical analysis.
In our study, we included a total of six participants, which was determined based on the availability of athletes who met the specific criteria for participation in goalball. While we recognize that this sample size is relatively small, it was a practical decision given the constraints of recruiting visually impaired athletes who are actively engaged in competitive goalball. It was explained in this paragraph: “The sampling employed in this study was purposeful, rendering the calculation of the sample size inappropriate. Instead, probabilistic sampling techniques were utilized for this calculation. While this approach imposes limitations on external validity and the generalizability of results, it does not compromise the internal validity of the findings.”
We acknowledge that the small sample size may limit the generalizability of our results to the broader population of goalball athletes. In future studies, we aim to recruit a larger and more diverse sample to strengthen our findings and provide more robust conclusions. Thank you again for your insightful comment.
- Why was the heart rate assessed in GPS correlation?
Answer: Thank you for your insightful question regarding the assessment of heart rate in our study. The heart rate band used was connected to the GPS system solely for data recording purposes; however, it is important to clarify that we did not utilize the GPS data in our analysis. The heart rate was monitored to gain insights into the physiological responses of the athletes during gameplay, allowing us to evaluate the effort and physical stress they experienced.
We appreciate your comment, as it made us realize that the phrase "Furthermore, the integration of heart rate monitoring with GPS technology facilitated a comprehensive understanding of the relationship between physiological responses and performance metrics, enhancing the overall analysis of the study" was nonsensical in the context of our study and has been removed. We hope this clarification is helpful, and we thank you for your guidance in improving the clarity of our research.
Results
- The correlation in results are random. Place scatterplots of points for these correlations. You can deleted Fig 2.
Answer: We have investigated the differences in the number of impacts in each zone between the first and second halves of the study. In Figure 1, each point represents an individual subject, and most points either ascend or descend, indicating a trend. This trend is corroborated by the results presented in Table 2, which show statistically significant differences in the total number of impacts, as well as in zones 2 and 5, when comparing the first and second halves. Therefore, we consider the information provided in Figure 1 to be valuable. Additionally, there is no Figure 2 in our manuscript.
The results comparing the first and second halves are more interesting, but they also do not contribute anything and do not correspond to the set goal.
Discussion
- You write that players experiencing accumulated fatigue will adjust their defense strategies to avoid long-term exposure, but this is not your research, just speculation.
Answer: some parts of the discussion have been deleted (including “and suggest a shift in the game dynamics” or “and it might suggest a shift in the nature of the game as it progresses, possibly with a different tactical focus or a reduction in impact intensity”) and “might” has been added to avoid stating ideas that have not been researched.
The second paragraph has been re-written and can be read as “Despite the narrow confidence intervals for the correlation between impacts in zones 2 and 4 in the second half, which suggest high precision and reliability in these estimates, there are wide confidence intervals for other correlations. Specifically, the correlations be-tween dynamic stress load and impacts in zone 6 in the first half, between the percentage of impacts in zone 1 in the first half and zone 6 in the second half, and between impacts in zone 3 in the first half and dynamic stress load in the second half, indicate significant un-certainty in the strength of these relationships, despite their statistical significance”
You further write that this tactical adaptation may explain the relationship between stronger shots in the first part of the match and weak or moderate shots in the later stages of the match, but this is not confirmed by the results of significant correlations in the first half - there is only 1 and it is rather accidental. In general, it is impossible to make rational generalizations based on the number of 6 people.
- Again, you write that This tactical adaptation may explain the relationship between stronger hitting in the early part of the match and weak or moderate hitting in the later stages of the match. Decide what the important factor was.
Answer: this sentence has been replaced by “Therefore, stronger impacts have been shown during the first part while low to moderate impacts in the later stages of the match.” To avoid confusion.
- Many parts of the discussion is not about research results. You can delete it. Your results do not address tactical adaptation and cannot explain the relationship between stronger shots in the first part of the match and weak or moderate shots later in the match.
Some parts of the discussion, such as “and suggest a shift in the game dynamics” or “and it might suggest a shift in the nature of the game as it progresses, possibly with a different tactical focus or a reduction in impact intensity” have been deleted.
- The summary does not indicate what you have achieved because you have not studied stress management.
Thank you for your valuable feedback. We acknowledge that the study did not explicitly investigate stress management strategies, so this has been removed from conclusions. Instead, our primary focus was on examining the relationship between trunk impacts and dynamic stress load in goalball athletes. We have revised the summary and conclusion sections to more accurately reflect the scope and findings of the study.

Reviewer 3 Report
Comments and Suggestions for Authors
The article needs a rethink and a more precise description.
The title should be changed. This is a case study
Introduction
- „However, there is a lack of evidence about performance factors and how to assess them on goalball (Petrigna et al., 2020)” - second paragraph not citing correctly
- The goal of scientific research is to obtain new knowledge. There is a lack of research questions or hypotheses
Materials and Methods
Participants
- “Six goalball players (32 ± 8 years old; 4 males) participated in this study.” How many players were examined? 6 or 4 ?
Statistical Analysis
- “Effect size (ES) was calculated as 𝐸𝑆=𝑊/√𝑛 using… “ Explain the designations W and n ?
Results
- “The study included a total of six participants who played a game of goalball. These players had a mean weight of 76.5 ± 20.1 kg and a height of 172 ± 8 cm. On average, they had 10 ± 2 years of experience in playing goalball.”
These are not the results
Conclusions
- No clearly indicated new discoveries
References
- Very poor literature review. References not tailored to meet editorial requirements
Author Response
Thank you for your thorough review of our manuscript. We greatly appreciate the time and effort you have dedicated to providing valuable feedback. We have made every effort to address all of your comments and suggestions in the revised version. Your insights have significantly contributed to enhancing the clarity and quality of our work.
The title should be changed. This is a case study
Answer: Thank you for your valuable feedback regarding the title of our manuscript. However, we think that this research is not a case study but rather a cross-sectional study. A case study typically involves an in-depth examination of a single individual or a small group, focusing on specific instances or phenomena. In contrast, our study assessed the impacts sustained by multiple goalball players during a national league match, analyzing the relationship between trunk impacts and dynamic stress load across the entire group. This design allows for broader insights into the physical demands and performance factors affecting visually impaired athletes in goalball.
We appreciate your suggestion and thank you once again for your constructive comments.
Introduction
- However, there is a lack of evidence about performance factors and how to assess them on goalball (Petrigna et al., 2020)” - second paragraph not citing correctly
Answer: Thank you for your insightful comment regarding the citation in the second paragraph. We appreciate your attention to detail. We have corrected this mistake.
- The goal of scientific research is to obtain new knowledge. There is a lack of research questions or hypotheses
Answer: Thank you for your valuable feedback regarding the clarity of our research questions and hypotheses. In response to your comment, we have added the following paragraph to the manuscript, at the end of the introduction:
"The hypothesis of this study was that dynamic stress load is positively correlated with the number and intensity of trunk impacts during goalball gameplay. Additionally, it was hypothesized that trunk impact dynamics change between the first and second halves of a goalball match."
We believe that this addition clearly articulates our research hypotheses and provides a solid foundation for understanding the objectives of our study. We appreciate your suggestion, as it has helped us enhance the clarity and focus of our research.
Materials and Methods
Participants
- “Six goalball players (32 ± 8 years old; 4 males) participated in this study.” How many players were examined? 6 or 4 ?
Answer: Thank you for your insightful comment regarding the participant demographics in our study. To clarify, we have revised the text to specify that a total of six goalball players participated in the study, which included four males and two females. This addition ensures that the participant composition is clear and accurately represented.
Statistical Analysis
- “Effect size (ES) was calculated as ??=?/√? using… “ Explain the designations W and n ?
Answer: Thank you for your thoughtful question regarding the calculation of effect size (ES) in our study. To clarify:
W: This designation refers to the Wilcoxon signed-rank test statistic. The Wilcoxon signed-rank test is a non-parametric statistical test used to compare two related samples or repeated measurements on a single sample. It assesses whether their population mean ranks differ, making it particularly useful for our analysis of paired data in the context of goalball gameplay
n: This represents the number of observations or pairs of data points in the sample. In the context of this study, it refers to the number of goalball athletes participating in the study or the number of impact measurements taken during the gameplay analysis.
We appreciate your inquiry, as it allows us to provide a clearer understanding of our statistical methods. An explanation about that has been added.
Results
- “The study included a total of six participants who played a game of goalball. These players had a mean weight of 76.5 ± 20.1 kg and a height of 172 ± 8 cm. On average, they had 10 ± 2 years of experience in playing goalball.”. These are not the results
Answer: Thank you for your comment. The paragraph you referred to, which included participant demographics and experience, has been removed from the results section and will be appropriately placed in the methodology section to provide context on the sample population used in the study. This ensures clarity and proper organization of the study's findings and background information.
Conclusions
- No clearly indicated new discoveries
Answer: Thank you for your valuable feedback. Based on your suggestion, we have revised the conclusion to emphasize the novel discoveries and their significance. The revised conclusion now reads as follows: "In conclusion, the present study highlights the importance of understanding the interplay between trunk impacts and dynamic stress load in goalball athletes, especially regarding defensive aspects. The study found significant positive correlations between dynamic stress load and impacts in zone 6 in the first half, and impacts in zones 2 and 4 in the second half. These findings represent a new discovery in the context of goalball performance analysis.
Moving forward, future research should consider exploring targeted training interventions aimed at reducing fatigue and optimizing performance in goalball athletes. Implementing strength and conditioning programs focused on core stability, balance, and endurance may mitigate central fatigue, minimize injuries, and enhance players' resilience to high-force impacts."
We believe these changes address your concern by clearly highlighting the novel findings of our study and their implications for future research and practice. Thank you for helping us improve our manuscript.
References
- Very poor literature review. References not tailored to meet editorial requirements
Answer: We consider that the references are in accordance with the journal's format. The limited number of references is due to the lack of scientific evidence on this topic.

Reviewer 4 Report
Comments and Suggestions for Authors
Greetings to the Authors who undertook the study entitled "Quantifying Trunk Impact Dynamics through inertial sensors in Goalball Athletes and its relationship with Workload", which places science at the service of improving the training conditions of such special athletes. However, I would like to call your attention for some issues I think deserve a reanalysis from you, which I detail in the report attached. I hope you see the comments constructively, insofar they were thought to add my perspective towards some issues you might have not considered before.
Best regards,

Author Response
Thank you for your thorough review of our manuscript. We greatly appreciate the time and effort you have dedicated to providing valuable feedback. We have made every effort to address all of your comments and suggestions in the revised version. Your insights have significantly contributed to enhancing the clarity and quality of our work.
- "...there is a lack of evidence about performance factors and how to assess them on goalball (Petrigna et al., 2020)." Why this indirect citation is not numbered like the others throughout the text?
Answer: This indirect citation should have been numbered consistently with the other references throughout the text. The mistake has been acknowledged and corrected to ensure that all citations follow the same referencing style for clarity and coherence. Thank you for bringing this to our attention.
- "..., Cursiol et al. [7] showed the occurrence of central fatigue on the second and third games after a first game, ..." It seems important to highlight, for those who might think 3 games in a roll is an uncommon situation whatever sports we are talking about, that according the Goalball Rules and Regulations for the 2022-2024 period, issued by the International Blind Sports Federation (p. 47, section 51), a goalball team can play up to 3 games in the same day. Explain that way will allow readers to realize why fatigue may be a challenging situation for goalball athletes.
Answer: Thank you for your insightful comment regarding the occurrence of central fatigue as reported by Cursiol et al. We appreciate your suggestion to clarify the context of playing multiple matches in goalball. To address this, we will include a statement in the manuscript highlighting that, according to the Rules and Regulations of Goalball for the 2022-2024 period, issued by the International Blind Sports Federation (p. 47, section 51), a goalball team is permitted to compete in up to three matches in a single day. This information is crucial as it illustrates the potential for significant physical and mental fatigue that athletes may experience during competitions, thereby emphasizing the relevance of the findings related to central fatigue. Thank you once again for your valuable feedback.
- Besides, it also seems important to provide here or further, for instance, in the next paragraph, a more detailed explanation about the concept of central fatigue as a reduced ability of the central nervous system to fully activate a fatigued muscle during voluntary contractions (Gandevia 1992) and has been found to develop in several muscles and during various types of contractions (Bigland-Ritchie et al. 1986; Lloyd et al. 1991; McKenzie et al. 1992; Loscher et al. 1995).
Answer: Thank you for your valuable feedback. In response to your suggestion, we have added a detailed explanation of the concept of central fatigue to the introduction. The added paragraph is as follows:
"Research about central fatigue showed that it occurs when the central nervous system struggles to fully activate muscles during prolonged efforts. While muscles like the quadriceps can maintain activation, others, such as the soleus, may experience significant central fatigue (Fatigue of Intermittent Submaximal Contractions). Central fatigue refers to the progressive reduction in voluntary muscle activation during exercise, leading to decreased muscle function, impaired reaction times, and decision-making, significantly impacting athletic performance, particularly in team sports, where athletes often engage in prolonged periods of high-intensity activity."
We hope this addition addresses your concern by providing a comprehensive explanation of central fatigue and its implications for athletic performance.
- "This study aims to utilize inertial technology to quantify the dynamics of trunk impacts in Goalball athletes and investigate their correlation with central fatigue and dynamic stress load." The quantifying of those dynamics of trunk impact will be assessed only when the players are in defense phase or throughout the whole game without considering the trunk impact context? The issue emerges from the fact the third paragraph of introduction underscore the importance of core stability, which includes the trunk zone of the body, as it absorbs the force generated by its impact over the floor. Besides, your conclusion clearly states "...understanding the interplay between trunk impacts and dynamic stress load in goalball athletes, especially regarding defensive aspects and stress load management, ...", and so this issue should be clarified from the beginning.
Answer: Thank you for your insightful feedback. In response to your concern, we have clarified the scope of our impact quantification by specifying that it will be assessed during the defensive phase, as trunk impacts rarely occur during the offensive phase. The revised paragraph in the introduction now reads:
"The quantification of impacts during the defensive phase on Goalball players and their association with stress load and fatigue holds significant potential for optimizing the performance of visually impaired athletes."
We hope this addition addresses your concern by clearly delineating the context in which trunk impacts will be assessed.
- "Informed consent form was signed before the match and procedures of the study were explained to the participants, with the collaboration of the researchers and ONCE collaborators." The first appearance of an acronym should be preceded by what it represents word-by-word and not ahead.
Answer: Thank you for your feedback. The error has been corrected. The definition of the acronym has been included the first time it appears in the text.
- "Six goalball players (32 ± 8 years old; 4 males) participated in this study." Why age and gender were not taken into the analysis? Though the reduced sample size, it would be possible, at least, to descriptively compare men and women, allowing readers to realize if this variable may or may not impact DSL during a goalball game.
Answer: Thank you for your feedback. We considered your suggestion; however, we believe that the sample size is insufficient to provide a relevant analysis of gender differences. Additionally, the age of the participants was very similar, which we believe would not yield significant insights into its impact on DSL during a goalball game. Therefore, we chose not to include these variables in the analysis.
- "Individuals with neuromuscular or cardiovascular conditions within the three months preceding the study were not included." How this situation was ensured? Medical records were consulted?
Answer: Thank you for your question. Medical records were not consulted as we do not have the legal authority to do so. However, potential participants were asked about their medical history during the explanation of the research, and we ensured this exclusion criterion was met during the informed consent process.
- "The impacts were sorted into six zones: zone 1 (impacts between 3 and 4 G-force), zone 2 (impacts between 4 and 5 G-force), zone 3 (impacts between 5 and 6 G-force), zone 4 (impacts between 6 and 7 G-force), zone 5 (impacts between 7 and 8 G-force), and zone 6 (impacts between 8 and 9 G-force)." Perhaps the term "zone" does not exactly convey what is supposed to mean, which would be the strength athletes bodies hit the ground as the result of an intervention on the ball, right? Why not use "impact-level" instead?
Answer: Thank you for your suggestion. We've decided to change the terminology to better reflect the strength of impacts athletes experience as a result of an intervention on the ball. The zones have been renamed as follows:
Zone 1 (impacts between 3 and 4 G-force): Very Low Impact Level
Zone 2 (impacts between 4 and 5 G-force): Low Impact Level
Zone 3 (impacts between 5 and 6 G-force): Moderate Impact Level
Zone 4 (impacts between 6 and 7 G-force): High Impact Level
Zone 5 (impacts between 7 and 8 G-force): Very High Impact Level
Zone 6 (impacts between 8 and 9 G-force): Extreme Impact Level
This new naming convention aims to provide a clearer understanding of the impact levels experienced by athletes.
- "Impacts below 3 G-force and above 9 G-force were excluded from consideration." What was the rationale for this interval setting?
Answer: Thank you for your comment. The rationale for setting the interval between 3 G-force and 9 G-force is based on the nature and occurrence of impacts in sports. Impacts above 9 G-force are excluded because they are not typically encountered in sports and this is also the measurement limit; the GPS system applies a filter and does not calculate them. In fact, the maximum recorded impact is 8.3 G-force. Impacts below 3 G-force are excluded as well, as they likely pose minimal injury risk and are similar to the G-forces experienced in daily life.
- "This criterion categorized the correlation coefficients on a scale from ±0.001 to ±0.299 as Poor, ±0.300 to ±0.599 as Fair, ±0.600 to ±0.799 as Moderate, ±0.800 to ±0.999 as Very strong, and ±1 as Perfect." Based on what theoretical framework?
Answer: Thank you for your feedback. We have added a bibliographic reference to support the criterion used for categorizing the correlation coefficients.
- "..., notable effect sizes were observed for variables such as the total number of impacts and the maximum heart rate, ...". If the effect sizes were notable, why the variation of heart rate as a function of the total number of impacts was not represented through a table, facilitating readers perception of such effect?
Answer: Thank you for your comment. The notable effect size refers to the comparison between the first and second halves for both heart rate and the total number of impacts (the latter is already represented in graphs). However, it does not refer to the relationship between these variables; therefore, a table representing that relationship would not provide significant additional information.
- "Performance factors such as anthropometry, core stability, acoustic reaction time, neuromuscular control, and proprioception seem to influence the defensive role of players, potentially affecting both impact and the fatigue it induces..." If the literature points out anthropometry as a performance factor on athletes’ impact and fatigue, why the weight and height of your sample were not included into the analysis? Although it is a reduced sample, the variability observed - specially on body composition - could add important insights to what have already been found in the present study.
Answer: Thank you for your insightful comment. The focus of the article was on quantifying trunk impacts and their relationship with load, rather than on performance factors. Including weight and height in the analysis could not be appropriate due to the small sample size, which similarly affected other variables such as gender and age. The primary goal was to establish a clear understanding of the impacts experienced by the trunk, and incorporating additional performance factors would have required a larger, more diverse sample to yield meaningful results.
- "...the present study highlights the importance of understanding the interplay between trunk impacts and dynamic stress load in goalball athletes, especially regarding defensive aspects and stress load management, showing significant positive correlations between dynamic stress load and impacts in zone 6 in the first half, and impacts in zones 2 and 4 in the second half." I honestly think you could have so much more to take from your study than just simply conclude by the existence of a correlation, if you had considered other analysis on your statistical drawing, as, for instance, a factor analysis that would allow you to quantify the contribution of variables like age, gender, weight, height and/or game period (1st or 2nd half) to the amount of dynamic stress load a goalball athlete is undergone during a game.
Answer: Thank you for your comment. As mentioned in previous responses, the sample size in our study was not sufficient to perform the kind of detailed factor analysis you suggested. We specifically chose correlation analysis because it allows us to analyze the large amount of data reported by the devices. Although we only had six participants, we collected a substantial amount of data from each, covering an entire match. This approach enabled us to draw meaningful conclusions about the relationship between trunk impacts and dynamic stress load without the need for a larger sample.

Round 2
Reviewer 1 Report
Comments and Suggestions for Authors
Dear Authors,
I appreciate the effort to address the raised questions and improve the work. The writing has been significantly enhanced. However, I don´t understand why images and diagrams were not included. Since this is an experimental work, visual aids are important for readers. I know, but I regret that.
Author Response
Thank you for your observation and for acknowledging the improvements in the writing. We fully agree that including images and diagrams would have been beneficial, especially in an experimental work where visual aids can greatly enhance the reader's understanding. Unfortunately, no photographs were taken during the experiments, which would have been very appropriate to include.
Given the simplicity of the methodology, we prioritized comparative graphs between the first and second phases of the experiment over diagrams. However, we recognize the potential value of the reviewer's suggestions and appreciate the insight.
Reviewer 4 Report
Comments and Suggestions for Authors
Dear Authors, this second version of your manuscript contains most of the corrections made to the first version, although other questions have remained unanswered, which I understand, as accepting them would imply a profound redesign of your work and, consequently, a significant delay in its schedule relating to the publication of this work. However, I leave here a suggestion so that the comments not included in this manuscript can be the subject of a second publication of yours on this topic.
Best regards,

Author Response
Thank you for your thorough review and for recognizing the improvements made in this second version of our manuscript. We appreciate your understanding of the challenges involved in addressing some of the more substantial suggestions.
We find your suggestion to consider the unanswered comments for a potential second publication highly valuable. Your insights have provided us with ideas for future research, and we will certainly explore these aspects in greater depth in our subsequent work on this topic.
Thank you once again for your constructive feedback and for helping us enhance the quality of our research.